# The Innovative Informatics Approaches of High-Throughput Technologies in Livestock: Spearheading the Sustainability and Resiliency of Agrigenomics Research

**DOI:** 10.3390/life12111893

**Published:** 2022-11-15

**Authors:** Godagama Gamaarachchige Dinesh Suminda, Mrinmoy Ghosh, Young-Ok Son

**Affiliations:** 1Interdisciplinary Graduate Program in Advanced Convergence Technology and Science, Jeju National University, Jeju-si 63243, Republic of Korea; 2Department of Biotechnology, School of Bio, Chemical and Processing Engineering (SBCE), Kalasalingam Academy of Research and Educational, Krishnankoil, Virudhunagar 626126, India; 3Department of Animal Biotechnology, Faculty of Biotechnology, College of Applied Life Sciences, Jeju National University, Jeju-si 63243, Republic of Korea; 4Bio-Health Materials Core-Facility Center, Jeju National University, Jeju-si 63243, Republic of Korea; 5Practical Translational Research Center, Jeju National University, Jeju-si 63243, Republic of Korea

**Keywords:** agrigenomics, animal genetics, genome database, high-throughput technologies, infectious diseases

## Abstract

For more than a decade, next-generation sequencing (NGS) has been emerging as the mainstay of agrigenomics research. High-throughput technologies have made it feasible to facilitate research at the scale and cost required for using this data in livestock research. Scale frameworks of sequencing for agricultural and livestock improvement, management, and conservation are partly attributable to innovative informatics methodologies and advancements in sequencing practices. Genome-wide sequence-based investigations are often conducted worldwide, and several databases have been created to discover the connections between worldwide scientific accomplishments. Such studies are beginning to provide revolutionary insights into a new era of genomic prediction and selection capabilities of various domesticated livestock species. In this concise review, we provide selected examples of the current state of sequencing methods, many of which are already being used in animal genomic studies, and summarize the state of the positive attributes of genome-based research for cattle (*Bos taurus*), sheep (*Ovis aries*), pigs (*Sus scrofa domesticus*), horses (*Equus caballus*), chickens (*Gallus gallus domesticus*), and ducks (*Anas platyrhyncos*). This review also emphasizes the advantageous features of sequencing technologies in monitoring and detecting infectious zoonotic diseases. In the coming years, the continued advancement of sequencing technologies in livestock agrigenomics will significantly influence the sustained momentum toward regulatory approaches that encourage innovation to ensure continued access to a safe, abundant, and affordable food supplies for future generations.

## 1. Introduction

Due to the incentives for developing quantitative theories and methodologies, high-throughput next-generation sequencing (HT-NGS) technologies have become more accessible. They are now employed in numerous biological science sectors [1]. The large-scale genome databases and sophisticated bioinformatics tools can expand new avenues of research with a wide range of applications including, but not limited to, chromatin immunoprecipitation coupled with DNA microarray (ChIP-chip) or sequencing (ChIP-seq), RNA sequencing (RNA-seq), whole-genome genotyping, de novo genome assembling and reassembling, genome-wide structural variation, mutation detection, and carrier sequencing [2,3]. The recent development of high-throughput ‘benchtop’ sequencers empowers laboratories to sequence. These tremendously significant advancements are the direct consequence of an ingenious interplay of chemistry, engineering, software, and molecular biology to produce, process, and evaluate large datasets generated by comparative genomics.

As the new genomics era matures, the development of novel bioinformatic algorithms has facilitated the NGS technologies and has now become the backbone of agrigenomics research [4]. The worldwide next-generation sequencing services market was worth USD 1.03 billion in 2021 and is predicted to rise at an 18.3% compound annual growth rate (CAGR) from 2022 to 2030. Considering that NGS is largely utilized for research, universities and other research institutions were reported to own the most significant revenue share of more than 50.0% in 2021 [5]. The new sequencing platforms, cloud computing, and sequence analysis tools have rapidly transformed microbiological research by allowing applications in clinical diagnostics, drug discovery, public health, microbiome research, antimicrobial resistance studies, and industrial and environmental microbiology. Research organizations are using sequencing services to improve project outcomes and obtain in-depth insights into disease mechanisms. The advancement of genome sequencing has improved the scientific comprehension of agrigenomics that underlies its economic features and has allowed us to anticipate the phenotypes related to yield efficiency and animal health.

Animal agriculture must become more robust and adaptive to sustain global food security and ensure human health [6]. However, in recent decades emerging infectious illnesses connected with domestic and companion animals, such as foot and mouth disease, bovine spongiform encephalopathy, avian influenza, and African swine fever, have considerably increased, causing significant threats to human and animal health. The direct cost of zoonotic diseases is estimated to be more than USD 20 billion, with indirect costs in affected economies totaling more than USD 200 billion [7]. Thus, despite the growing global demand for safe and sustainable animal products, a lack of general animal husbandry knowledge and the emerging livestock diseases can place domestic animals at a higher risk of acquiring zoonotic diseases [8,9].

Recent studies using high-throughput sequencing (HTS) have provided unique insights into the inference of transmission pathways during global pandemics and localized outbreaks and the pathogens’ evolution over colonization and infection [1,2]. The complexities of host–pathogen interactions dictate the progression and outcome of infectious illness. The association of HTS with what are often called the hit sequences (HITS), such as, Transposon sequencing (Tn-Seq), transposon-directed insertion-site sequencing (TraDIS), insertion sequencing (INSeq), and identified sequences, has simplified the screening of libraries containing hundreds of thousands of infectious pathogens [3].

Some of the microbial genome sequences currently available in the public databases and the genomic data are questioned for their accuracy, completeness, authenticity, and traceability since they could have been generated by researchers using unauthenticated cultures and earlier sequencing and analysis techniques. The underlying issues are further rendered by the lack of standardized methodologies for best practices in reference genome sequencing and assembly [10]. The researchers must have access to reliable genetic information that can be traced back to verified, fully described materials with known and reliable sources. In the current, concise review, we summarize selected studies that have applied HTS in domestic animal reference sequences to answer important questions regarding the success of sequencing (HT-NGS) technologies in livestock agrigenomics research. We provide an overview of some of the insights into the historical perspectives of sequencing technologies and their impact on the genetic potential of economically important farm animals, such as cattle, swine, chickens, horses, ducks, and sheep. Furthermore, the various bioinformatics resources to support genome research in selected animals are reviewed. We also summarize the current role of sequencing methods and platforms for monitoring, detecting, diagnosing, and controlling zoonotic infectious diseases.

## 2. The Era of the Development of Sequencing Technologies 

The period of current sequencing technology began in 1959, with the discovery of the bacteriophage ϕX174 genome, when Sinsheimer purified the first DNA molecule to homogeneity [11]. However, the discovery of type II restriction enzymes by Hamilton Smith and colleagues (in 1970–1972) substantially changed the approach to modern genetics, and DNA sequencing could not have occurred without this discovery [12] (Figure 1). The first whole genome of a bacteriophage (MS2) was announced, and in 1972 Sanger’s “plus and minus” sequencing method was identified as a crucial transition technology leading to modern methods [13]. Thus, current DNA sequencing began in 1977 with the invention of the chemical approach of Maxam and Gilbert and the dideoxy method of Sanger, Nicklen, and Coulson [14].

### 2.1. The First Generation of Sequencing Technologies

The throughput sequencing methods can be divided into three generations of sequencing technologies: post-monopoly era sequencing platforms, second-generation, and third- and fourth-generation sequencing platforms [15]. In 1986, near the end of the capillary electrophoresis technique sequencing era, further development resulted in an automated fluorescent technique to sequence a genome region. The primary technology in the “first generation” of automated DNA sequencing was reported using Applied Biosystems (ABI) fluorescent sequencing [16]. Efforts are being made to improve the sequencing techniques, allowing for the development of increasingly automated DNA-sequencing equipment with fluorometric detection and enhanced sensing employing capillary-based electrophoresis. The Welcome Trust and Medical Research Council integrated a global public effort to sequence the human genome in the Human Genome Project. The project initially began in 1990 and was completed in 2003. Although the Sanger technique was used to sequence the first 3.0 billion bp of the human genome (released in 2000), the human reference genome has only covered the euchromatic part of the genome, rendering crucial heterochromatic regions incomplete. The Genome Reference Consortium (GRC) released the current human reference genome in 2013 and most recently updated it in 2019 (GRCh38.p13). This reference has evolved over the past 20 years and can be attributed to the Human Genome Project [17].

Several unique sequencing technologies were developed two decades after the advent of electrophoretic techniques for DNA sequencing. The words “next-generation” and/or “massively parallel” DNA sequencing is used to refer to the DNA HTS technologies that can sequence a large number of distinct DNA sequences in a single reaction. Sanger-based “topdown” techniques need to characterize large clones by low-resolution mapping in microtiter plate wells, whereas massively parallel approaches do not. The main premise of the NGS approaches is based on the DNA ligase covalently attaching the synthetic DNA adapters to each of the targeted fragment ends and the in situ amplification on a solid surface. The Solexa technology was developed in 1998, and the 454 Life Science in 2000. However, the GS20 454 sequencing platform debuted in 2005 and was the first non-Sanger-based commercialized technology [18]. The Roche 454, the first commercial NGS platform, employed large-scale parallel pyrosequencing chemistry to identify base pair sequences with higher throughput and lower sequencing costs per base than the Sanger sequencing [19].

### 2.2. The Second Generation of Sequencing Technologies

The NGS methodology has been used in many fields, including transcriptome analysis, de novo assembly, genotyping, targeted and whole genome sequencing, and the detection of SNPs, copy number variation, exome, protein–protein interactions, and genome methylation. The Roche 454 genome technology, which is based on Melamede’s sequencing-by-synthesis (SBS) theory (1985), was the first next-generation system to be commercially viable and uses pyrophosphate to identify the pyrophosphate generated during DNA synthesis. The Roche 454 GS system was initially released in 2005, and in 2008 it was updated to the Roche GS-FLX 454 Titanium system. The GS-run processor and the additional work in 2009 streamlined the library preparation and data processing. Roche employed a GS FLX+ sequencer capable of reading 400–600 million base pairs each run with maximum pair-read lengths of 1000, however, Roche 454 was phased out in 2016 [20].

The SOLiD platform, developed by Harvard Medical School and the Howard Hughes Medical Institute, was commercially released by ABI in October 2007 and generated 4 Gb of sequencing data within the six days of running [12]. This sequencing system employs the sequencing-by-ligation method of oligonucleotide ligation and detection. The SOLiD sequence is based on color coding, which is decoded to produce the basic sequence. However, incorrect color coding might result in decoding errors. Balasubramanian and Klenerman utilized fluorescently tagged nucleotides in the middle of the 1990s to observe a single polymerase molecule migrate [12]. In June 2006, the first Solexa sequencer was launched, and Illumina entered the industry in 2006, bought Solexa in 2007, and gradually progressed the NGS industry [16]. A paired-end module for the sequencer with new optics and camera components was included in the Genome Analyzer II in 2008 as a result of further advancements in the Illumina method [12].

Ion Torrent Systems Inc. (Gilford, CT, USA), in 2010, invented the first commercial sequencing method that did not rely on dye-labeled oligonucleotides and expensive optics. It monitors H ions generated during base incorporation and is specifically suited to amplicon sequencing. Despite its benefits, Ion Torrent’s read accuracy remains a major challenge. The high rate of mistakes induced by the noisy sequencer signal is translated into a nucleotide sequence. Furthermore, the signal decays over time, resulting in a drop in the signal-to-noise ratio [21]. Polonator, a polony sequencing machine, was invented by Dr. George Church’s group at Harvard Medical School in 2009. Polony sequencing, a non-electrophoretic sequencing technology, can read millions of immobilized DNA sequences simultaneously at a lower cost per nucleotide than conventional Sanger sequencing. The fundamental limitation of this method is the non-uniform amplification, which results in a decreased sequencing accuracy and a read length of just 26 bp [22]. The second-generation 454 GS-FLX, Illumina, and SOLiD sequencing systems are not sensitive enough to detect the individual single-molecule template extensions, whereas “third- or fourth-generation sequencers” are “single molecular”-type sequencers, such as the Heliscope, PacBio, and Oxford nanopore sequencers, which do not require pre-amplification steps and are more sensitive and precise.

### 2.3. Third-Generation Sequencing Platforms

Third-generation sequencing systems do not include an amplification step during library creation. They allow single-molecule sequencing with average read lengths reaching 6–8 kbp and maximum read lengths exceeding 30–150 kbp. Based on real-time imaging, the SMRT sequencing technology parallelizes data from a DNA polymerase and conducts uninterrupted template-directed synthesis. By emphasizing length, it breaks out of the existing short-read HTS instruments. In 2011, Pacific Biosciences made SMRT sequencing commercially available [23]. PacBio technology yields read durations ranging from 1000 to 3000 bp on average [12].

Illumina technology employs DNA colony sequencing, which is based on reversible dye terminator sequencing via synthesis chemistry. The Illumina sequencing-by-synthesis method is the most extensively used NGS technology because it provides precise read alignment and improved indel identification [19]. Early in 2010, Illumina introduced HiSeq 2000, and the continued research on cutting-edge flow cells for Illumina HiSeq technology led to the numerous novel sequencing platforms introduced from 2011 to 2018. Illumina has produced popular sequencing systems, including MiSeq, HiSeq, and NovaSeq [24]. Current NGS methods are at least 100 times quicker than traditional Sanger sequencing. Using NGS, complete genome sequences may be retrieved, providing fast and comprehensive information [25]. As a result, NGS technology is frequently employed to monitor gene expression across an organism’s genome.

Further development of HT next-generation platforms such as the GeneReader NGS technique, The 10X Genomics platform, The SeqStudio^TM^ Genetic analyzer, the Bionano Saphy^TM^ genomics platform, the fluorescence resonance energy transfer based GnuBio platform (Bio-Rad, Hercules, CA, USA), GenapSys, NanoString Technologies, an electron microscopy-based Electron Optica system and Firefly (Illumina, San Diego, CA, USA), nanopore sequencing by Genia (Roche, Basel, Switzerland),can revolutionize biological science through the ability to sequence more samples at higher depths, producing more insightful data in less time and at a lower cost per sample [26].

### 2.4. Fourth-Generation Sequencing Platforms

Following the three generations, a new type of sequencer was recently developed, represented by the PacBio sequencer and Nanopore sequencer, known as fourth-generation sequencing [27]. Oxford Nanopore Technologies (ONT) introduced two new TGS systems, MinION, PromethION, and GridION, in 2012, enabling the direct electronic study of DNA, RNA, proteins, and single molecules. This method uses nanopores and an exonuclease-based “deconstruction sequencing” approach. In 2014–2015, the MinIONs were distributed to selected laboratories for beta testing. Nanopore technology can provide real-time sequencing of single molecules for as little as USD 25–40 per Gb of sequence data. The data processing is simpler than the short-read sequencers because alignment and assembly are more straightforward using nanopore technology. GridION has tested up to five MinION Flow Cells simultaneously; it is a simplified benchtop infrastructure. It is ideal for labs with various applications that require the benefits of nanopore sequencing, such as facile library preparation, real-time analysis, and lengthy reads. PromethION is meant for HT and employs the same chemistry as MinION and GridION, which are intended for real-time usage. However, based on the number of samples, it has a high fidelity for DNA and RNA sequencing. It is a rapid sequencing method, and nanopore technology may represent the future of sequencing.

High-throughput approaches have tremendously aided research in obtaining genomic information for various species. The NGS platform’s current version supports directed readings and pathogen detection [28]. Several zoonotic pathogens are detected using the Illumina NGS platform. For example, the Illumina HiScan and MiSeq technologies have been utilized to broadly detect viral quasi species in the capsid gene area as evidence of positive selection allowing cell-tropism [29]. The ngs.plot algorithm visualizes the enrichment patterns of DNA-interacting proteins in functionally essential locations using NGS data; therefore, it is a helpful tool for bridging the gap between massive datasets and functionally important genomic information [30].

## 3. The Perspective of Domestic Animal Reference Sequences

The numerous domestic livestock species’ genomes, including those of chickens, pigs, cattle, sheep, and horses, have recently been partially or entirely sequenced (Table 1; Figure 2). The Red Junglefowl (RJF) chicken genome sequence was the first to be sequenced. The chicken genome’s initial draft was generated using an assembly with 6.6-fold whole-genome shotgun coverage. The Bovine Genome Sequencing and Analysis Consortium published the Taurine cow genome sequence in April 2009. This preliminary assembly identified around 22,000 genes and 14,345 orthologs shared by seven mammalian species. The first draft (98% complete) of the pig genome (*Sus scrofa*) constructed through global collaborative efforts has been made public. The diploid pig genome is about 2.7 109 kb long and comprises 38 chromosomes (including meta- and acrocentric ones). In 2010, the interim assembly version OARv2.0 for sheep was released to discover genes linked with sheep productivity, quality, and disease features. The OARv3.0 was finalized in 2012, with details on chromosomal gaps. In brief, we have discussed the perspective of the development of genome research in cattle, pigs, chickens, sheep, and horses.

### 3.1. Insights into Cattle (Bos taurus) Genome Research

Cattle have a long-standing relationship with human civilization and are essential in agriculture and research as model animals. Approximately 1.5 billion cattle are raised annually worldwide. The global demand for beef in 2019 was 70 million tons, along with bovine dairy products [31]. Thus, cattle represent significant scientific opportunities and a vital economic resource. In 2009, the first complete sequence of the bovine genome was published. The Centre for Bioinformatics and Computational Biology at the University of Maryland published a whole-genome assembly of *B. taurus* (2.86 billion bp) as well as the UMD 3.1 *B. taurus* assembly [32].

The Bovine Genome Sequencing Project was undertaken owing to the unique nature of ruminants and their role as a critical protein source for humans. The bovine genome sequence and haplotype map has transformed the beef and dairy sectors [33,34]. Many linkage maps have since been built to identify the economically significant features of the bovine family because the linkage map is predicted to include 90% of the bovine genome [35].

In recent years, the map of the bovine genome has also advanced rapidly. Chromosomal maps and synteny also facilitate the detection of chromosomal conservation in other species, particularly those relevant for extrapolating data from mouse and human maps to cattle [36]. Radiation hybrid mapping is a useful approach for creating in-depth comparison maps of single chromosomes and whole genomes [37]. Whole-genome shotgun sequencing has been used to discover possible segmental duplications and compare them with publicly accessible bovine genome sequence assemblies [38].

### 3.2. The Decade of Swine (Sus scrofa) Genomic Research 

According to molecular genetic data, the domestic pig (*S. scrofa*) is a eutherian mammal that emerged some 20–30 million years ago and originated in Southeast Asia [39]. Pork provides a high-quality protein source that can offer a highly desirable eating experience and supplies ~35% of all meat production with increasing global demand [40]. The pig is essential in biomedical research because of its ability to create transgenic and knockout pigs using somatic nuclear cloning methods, resulting in various models for specific human diseases. It has been reported that 112 positions in porcine protein sequences have amino acids implicated in human disease [41]. Traditional selective breeding can take years to produce a pig with all the desired characteristics, whereas modification of the pig genome can provide the same results in much less time [42].

The Swine Genome Sequencing Consortium (SGSC) initiated a whole-genome sequencing study for pigs in early 2006. The Wellcome Sanger Institute sequenced the whole pig genome using clone-by-clone sequencing. More than 287 Mb of sequencing have been completed from 1660 accessioned clones used in the project [43]. Indeed, high-throughput sequencing technologies have greatly improved the study of bacterial populations colonizing the porcine gut. These results reveal more nonredundant microbial genes between humans and pigs than between humans and mice. Thus, pigs are a better animal model than mice owing to their considerable similarities with humans [44].

### 3.3. Genetic Assembly Research in Chickens (Gallus gallus) and Ducks (Anatidae)

Chickens are the most popular fowls worldwide across different cultures and geographical areas and play a significant role in the rural economy in most underdeveloped and developing countries. Native chickens and ducks are reared in over 90% of rural homes. They are an essential element of a balanced farming system and serve as a source of high-quality animal protein in rural dwellings [45].

The chicken (*G. gallus*) is a key model organism for understanding the evolutionary relationship between mammals and other vertebrates. Genetic studies in chickens date back to the start of the twentieth century. The chicken genome comprises 38 autosomes and one pair of sex chromosomes, with the female as the heterogametic sex [46]. A consensus linkage map of the chicken genome has been created using all available genotyping data and has dramatically improved comparative gene mapping. This map shows that substantial syntenic areas between the human and chicken genomes seem to be consistently conserved [47].

Ducks (Anatidae) evolved from the related turkey, chicken, and zebra finch approximately 90–100 million years ago and are now one of the most commercially significant waterfowl for meat, eggs, and feathers [48]. The duck is also one of the most common domesticated waterfowl. Advances in NGS technologies have enabled population-level comparative genomic research to uncover the unique genetic features in domestic animals, including ducks. For example, 15.56 million single nucleotide polymorphisms have been discovered in Korean native ducks [49]. Fluorescence in situ hybridization (FISH) and microarray analysis are reported as: (i) vital tools for detecting large genomic rearrangements; (ii) copy number variants (CNVs); (iii) gene gains/losses; and (iv) gene order in the macrochromosomes of birds. Comparative genomics analysis has been conducted in chicken and Peking duck macrochromosomes using FISH mapping and microarray analysis. The results revealed one interchromosomal and six intrachromosomal rearrangements between these two species [50].

### 3.4. Genome Architecture in Sheep (Ovis aries)

The typical role of sheep is to provide meat, milk, and fiber as globally valuable commodities [51]. Sheep meat typically accommodates 3% of global meat production, and its quality depends on muscle quality and nutritional characteristics [52]. The introduction of NGS technology has allowed the attainment of vast amounts of sequence information at a substantially reduced cost [53]. Domestic sheep have 54 diploid chromosomes, of which 26 pairs are autosomes, and two are sex chromosomes. The identification and functional annotation of genes governing the various qualities of interest in sheep is critical.

The second-generation genetic map of sheep was created using 519 markers, and the genotypic data were merged using the international and USDA mapping flocks [54]. The completed genome spanned 2.62 Gb and comprised 7157 scaffolds with an N50 of approximately 2 Mb [55]. The International Sheep Genomics Consortium is working towards sequencing the reference sheep genome. The availability of the sheep genome sequence has allowed the anticipation of the functions of noncoding RNAs. Large-scale cDNA sequencing, also known as RNA-seq, provides complete transcriptome identification, annotation, and quantification [56]. Improvements in livestock breeding and awareness of desirable genetic traits across diverse breeds have also ushered in a new age in sheep genomics. Thus, the animal breeding sector is directly benefiting from the constant technological breakthroughs in NGS [57].

### 3.5. Inslight in the Horse (Equus caballus) Genomics

Horses have played a vital role in agriculture, transport, industry, and sport since their domestication 6000 years ago. The first whole genome of the horse was released in 2009 [58]. Since 1995, the Antczak laboratory has been a significant participant in the international collaboration for the Horse Genome Project, a consortium of over 20 laboratories from more than 12 countries that have collaborated to produce various genetic and physical maps of the horse genome, culminating in the whole genome sequence [59].

The Eli and Edythe Broad Institute of the Massachusetts Institute of Technology and Harvard University in Cambridge performed the horse genome sequencing and assembly. Paired-end low-coverage whole genome shotgun (WGS) of 100,000 reads each were generated from seven horse breeds (Arabian, Andalusian, Akhal-Teke, Quarterhorse, Icelandic horse, Standardbred, and Thoroughbred). The WGS reads were placed uniquely on the Equus1.0 Thoroughbred assembly, and the SSAHA-SNP tool was used for detection. The horse genome comprises 64 chromosomes [60], and the validation rate for these SNPs is estimated to be approximately 95%. Whole-genome sequencing of the horse genome has provided knowledge of equine genetic diversity; it has revealed 5.7 million single-nucleotide variations and 0.8 million minor indel variants, and some detrimental recessive alleles. This knowledge may facilitate the control of harmful recessive alleles in horse breeding programs and increase horse fertility [61]. According to the comparative genome sequencing of a late Pleistocene horse and the present genomes of five domestic horse breeds, all the current horses, zebras, and donkeys descend from the Equus lineage.

A study identified 29 genomic sites in horse breeds that depart from neutrality and display low variations compared to those in Przewalski’s horse [62]. FISH has been used to create a second-generation whole-genome radiation hybrid, cytogenetic, and comparative map of the horse genome. This map includes 4103 markers for all 31 autosomes and the X chromosome pairs. The resulting integrated map provides the most detailed information on the physical and comparative structure of the equine genome. It is a tool for identifying genes that regulate the horse’s health, illness, and performance [63].

The genomic maps of a male wild horse and a male Mongolian horse were improved by sequencing their genomes using NGS technology [64]. An assessment of the genomes of 38 normal horses from 16 different breeds revealed 258 CNV sites. Identifying variations contributing to equine genetic disorders requires a thorough understanding of CNVs in normal horse populations within and between breeds and must be undertaken [65]. Equus species exhibit higher karyotypic diversity than other animals and have a wide range of diploid chromosome counts, ranging from 32 in the mountain zebra to 66 in Przewalski’s horse.

## 4. Databases and Online Resources

Global assessment of population genetic diversity and identification of genome areas under natural and artificial selection have been facilitated by NGS [66]. However, challenges concerning the storage, accessibility and efficient visualization of massive datasets remain. The need for bioinformatics resources to enable genomic research in farm animals is widely acknowledged [67,68]. Genomic databases have been created to offer current summaries on the state of genetic analysis in various farm and domestic animals, as well as experimental details and links (Table 2). Large-scale genomic databases and helpful bioinformatics programs can provide new areas of study with a broad range of applications [69]. The resource databases and accompanying technologies have been created to manage vast amounts of experimental data. Several of these systems are designed to meet the requirements of global partnerships. Indeed, continuous development is necessary to keep the integrity and usability of existing services, especially genome databases.

## 5. Outline of Zoonosis Infections

Identifying and analyzing host–pathogen interactions (HPI) and Protein–protein interactions (PPIs) are critical in studying infectious diseases. However, the databases of molecular interactions that are accessible need not feature numerous HPI and PPI data, particularly for host–pathogen systems in agriculture [89,90]. Based on surveillance data, the CDC reports that the majority of zoonotic illnesses (41.4%) are bacterial, followed by viral (37.7%), parasitic (18.3%), fungal (2%), and prionic (0.8%). The number of online databases and tools available for discovery, annotation analysis, and archiving microbiome data are shown in Table 3.

Zoonotic pathologic changes can be transmitted from an infected animal or human to an exposed host [91]. Viruses, bacteria, fungi, and parasites are among the pathogens that cause these illnesses [92]. They may spread to humans via food, blood transfusion, vectors in the air, or direct contact [93]. Moreover, 60% of emerging infectious diseases are reported to originate from zoonotic pathogens [94]. The Center for Disease Control and Prevention estimates that, apart from the United States of America, 48 million people worldwide get sick from dietary products and 128,000 are hospitalized, while 3000 die of foodborne diseases yearly [95].

Infectious diseases in cattle, swine, horses, sheep, chickens, and produce from chickens cause significant economic losses for the livestock industry. Outbreaks of zoonotic contagious illnesses or reverse zoonotic disease transmission (zooanthroponosis) in humans are produced by pathogen spillover (cross-species spillover), and areas where humans and animals interact regularly, are possible spillover areas [96]. The severe acute respiratory syndrome coronavirus 2 (SARS-CoV-2) has been debated as either a zoonotic disease or an emerging infectious disease [96]. The COVID-19 pandemic has brought to public attention that even the highly developed and most qualified healthcare networks worldwide collapse when confronting a novel viral infectious disease of zoonotic origin. Before the COVID-19 pandemic, African swine fever significantly impacted the global livestock industry [97]. Following that, the COVID-19 pandemic has substantially influenced human health and the economy. The impact of the pandemic has also jeopardized the sustainability of livestock and agri-based products, significantly affecting the quality of life and causing economic losses. At the same time, more than 150 enteric viruses now recognized as crucial to human and animal health are considered in genomic surveillance efforts to monitor and forecast the subsequent pandemic spillover. In order to minimize economic losses in cattle production, advanced procedures must be prepared. Public health care considerations must also be accommodated [98].

**Table 3 life-12-01893-t003:** Existing online databases for pathogen genome-based research (Accessed on 3 November 2022).

Agents	Category	Resource	Description	URL	References
Virus	Genome database	National Center for Biotechnology Information’s (NCBI’s) virus	The National Center for Biotechnology Information hosts the Virus Variation Resource, a valuation viral sequence data resource that contains modules for seven viral groups, including the influenza virus, Dengue virus, West Nile virus, Ebolavirus, MERS coronavirus, Rotavirus A, and Zika virus. Pipelines that scan recently made GenBank records, annotate genes and proteins, parse sample descriptors, and map them to controlled vocabulary support each module.	https://www.ncbi.nlm.nih.gov/labs/virus/vssi/#/ (accessed on 3 November 2022)	[99]
	Genome database researching tool	Hmmer database	HMMER searches sequence databases for sequence homologs and performs sequence alignments. It is intended to detect distant homologs as sensitively as possible, relying on the robustness of its underlying probability models.	http://hmmer.org/ (accessed on 3 November 2022)	[100,101,102]
	Virus discovery and annotation tool	Cenote-Taker 2	Cenote-Taker 2 was written in Bash, Perl, and Python. All scripts can be found on GitHub. This tool is a virus discovery and annotation tool available via the command line and graphical user interface with free computation access, employs highly sensitive models of hallmark virus genes to discover familiar or divergent viral sequences from user-input contigs. Furthermore, Cenote-Taker2 employs a versatile set of modules to automatically annotate the sequence features of contigs, providing more gene information than comparable tools. The BLAST and Hmmer databases created for this tool can be found on Zenodo.	https://github.com/mtisza1/Cenote-Taker2 (accessed on 3 November 2022) https://zenodo.org/record/4031657 (accessed on 3 November 2022)	[103]
	Viral genomes identification database	IMG/VR	The IMG/VR database contains the most comprehensive collection of viral sequences obtained from (meta)genomes. The IMG/VR V3 contains 18 373 cultivated and 2 314 329 uncultivated viral genomes (UViGs), nearly tripling the total number of sequences compared to the previous version. These were divided into 935 362 viral Operational Taxonomic Units (vOTUs), with 188 930 having two or more members.	https://img.jgi.doe.gov/cgi-bin/vr/main.cgi (accessed on 3 November 2022)	[104]
	Microbiome analysis resource	MGnify	It offers a free platform for assembling, analyzing, and archiving microbiome data derived from sequencing microbial populations found in specific environments. MGnify’s increased focus on metagenomic data assembly has resulted in a six-fold increase in the number of datasets assembled and analyzed. MGnify’s Notebook Server provides a no-installation Jupyter Lab environment for users to explore programmatic access to MGnify datasets using Python or R via the MGnifyR package.	https://www.ebi.ac.uk/metagenomics/ (accessed on 3 November 2022) https://shiny-portal.embl.de/shinyapps/app/06_mgnify-notebook-lab?jlpath=mgnify-examples/home.ipynb (accessed on 3 November 2022)	[105]
Bacteria	Microbial Genome and Microbiomes database	IMG/M	The system serves as a public resource for genome and metagenome dataset analysis and annotation in a comprehensive comparative context. The IMG web user interface includes a number of analytical and visualization tools for comparing isolate genomes and metagenomes in IMG.	https://img.jgi.doe.gov/cgi-bin/m/main.cgi (accessed on 3 November 2022) https://img.jgi.doe.gov/ (accessed on 3 November 2022)	[106]
	MetaGenome Gene Finding	MetaGeneMark/2	MetaGeneMark’s developers, GENE PROBE Inc., have created and refined algorithms for gene prediction in metagenomic sequences for over fifteen years. This website provides access to gene prediction in metagenomes by utilizing metagenome parameters and gene prediction. This same MetaGeneMark-2 plugin has been further optimized for gene discovery in anonymous metagenomic sequences. In comparison to MetaGeneMark, estimated to be 2.7%, MetaGeneMark-2 reduces nearly twice the rate of false negative predictions and missed genes. MetaGeneMark-2 is a C++ program, and all experiments and results are run and analyzed in Python. All scripts can be found on GitHub.	http://opal.biology.gatech.edu/GeneMark/ (accessed on 3 November 2022) http://exon.gatech.edu/meta_gmhmmp.cgi (accessed on 3 November 2022) https://github.com/gatech-genemark/MetaGeneMark-2 (accessed on 3 November 2022)	[107,108,109]
	Genome database	Ensembl Bacteria	Ensembl Bacteria is a genome browser for bacteria and archaea. These are from the International Nucleotide Sequence Database Collaboration, the European Nucleotide Archive at the EBI, GenBank at the NCBI, and the Japanese DNA Database. The Ensembl Genomes project, launched in 2009, enhanced the Ensembl project by utilizing the same visualization, interactive, and programming tools to provide users with access to genome data from a further five domains: protists, bacteria, metazoa, plants, and fungi.	https://bacteria.ensembl.org/index.html (accessed on 3 November 2022)	[110,111,112]
	Bacterial Isolate Genome Sequence Database	BIGSdb	EBIGSdb is software that collects and evaluates sequencing data for bacterial isolates. BIGSdb extends the MLST concept to genomic data, allowing for the creation of many loci and assigning alleles based on sequence definition databases. The program is distributed under the GNU General Public License, version 3. The most recent version of this document may be obtained at https://bigsdb.readthedocs.org/ (accessed on 3 November 2022)	https://bigsdb.readthedocs.io/en/latest/ (accessed on 3 November 2022)	[113]
Parasite	Malaria Genome database	UCSC Malaria	The UCSC Genome Browser is an online and downloadable genome browser created by the University of California, Santa Cruz’s Hughes Undergraduate Research Group, in collaboration with Prof. Manuel Ares Jr.’s laboratory. It combines the entire DNA sequences of multiple malaria parasite species (*Plasmodium* sp.) on a single screen, together with experimental data and found genes from the literature. Users may browse through the malaria parasite’s genome’s 14 chromosomes, insert their sequencing data and annotations, and compare results across species.	https://plasmodb.org/plasmo/app (accessed on 3 November 2022)	[114]
	Eukaryotic Pathogen, Vector and Host Informatics Resource	PlasmoDB/ VEuPathDB	The database includes more than 500 organisms, including invertebrate vectors, eukaryotic pathogens (protists and fungus), and relevant free-living or non-pathogenic species or hosts. VEuPathDB projects integrate >1700 pre-analyzed datasets (and related metadata) with extensive search capabilities, visualizations, and analysis tools in a graphical interface to provide researchers with access to Omics data and bioinformatic studies.	https://plasmodb.org/plasmo/app (accessed on 3 November 2022)	[115]
	Model Organism Database for Caenorhabditis elegans	WormBase Parasite	It was established in 2000 and offered each species at WormBase a dependable and recognizable user interface. Furthermore, the WormBase Parasite V WBPS17 assembles the reliable, current information about the genetics, genomes, and biology of nematode *Haemonchus contortus* an animal endoparasite infecting wild and domesticated ruminants (including sheep and goats) worldwide.	http://www.wormbase.org (accessed on 3 November 2022) https://parasite.wormbase.org/Haemonchus_contortus_prjeb506/Info/Index/ (accessed on 3 November 2022)	[116,117,118]
	Global Mammal Parasite Database version 2.0	GMPD	GMPD, a database of parasites of wild ungulates (artiodactyls and perissodactyls), carnivores, and primates, and is provided for download as complete flat files. The updated database contains over 24,000 entries from over 2700 literature sources. It included data on sampling method and sample size when obtainable, as well as “reported” and “corrected” binomials for each host and parasite species. Current higher taxonomies and data on transmission modes used by the majority of the parasite species in the database are also included.	parasites.nunn-lab.org (accessed on 3 November 2022)	[119]
Fungi	Saccharomyces Genome Database	SGD	The SGD project delivers the highest-quality manually curated information from peer-reviewed literature and algorithms like sequence similarity searches, which leads to extensive details on genome characteristics and gene relationships. Researchers have public access to these data through online sites that are built for ease of use.	http://www.yeastgenome.org (accessed on 3 November 2022)	[120,121,122]
Common database microbial agents	Genome database for Archaea, Bacteria, Eukarya, Viruses	GOLD v.8	It is a data management system that manually catalogs sequencing efforts from around the world and the supporting metadata. In GOLD, there were 387,480 different creatures divided throughout 305 different phyla and candidate phyla. The bulk of these organisms (88%) are bacteria, followed by eukaryotes (8.5%), viruses (2.5%), and archaea (1%).	https://gold.jgi.doe.gov/ (accessed on 3 November 2022)	[123]
	Metagenomics RAST server	MG-RAST	The MG-RAST server is an open-source comparative genomics system based on the SEED platform. Users can upload raw fasta sequence data; the sequences will be normalized and analyzed, and summaries will be generated automatically. The service offers multiple methods for accessing the various data kinds, such as phylogenetic and metabolic reconstructions, as well as the ability to compare the metabolism and annotations of one or more metagenomes and genomes.	https://www.mg-rast.org/ (accessed on 3 November 2022)	[124,125]
	Evolutionary Genealogy of Genes: Non-supervised Orthologous Groups	EggNOG Database	EggNOG is a publicly available database that analyzes thousands of genomes at once to determine orthology links between all of their genes. It included a significant upgrade to the underlying genome sets, which were enlarged to include 4445 representative bacteria and 168 archaea generated from 25 038 genomes, 477 eukaryotic species, and 2502 viral proteomes.	http://eggnog5.embl.de/#/app/home (accessed on 3 November 2022)	[126]

Potential zoonotic exposure upon contact with cattle or their products causes concern as approximately 15.4 million pounds of beef products are rejected/canceled annually [127]. Bovine zoonoses, anthrax, brucellosis, cryptosporidiosis, dermatophilosis, Escherichia coli, giardiasis, leptospirosis, listeriosis, pseudo cow pox, Q fever, rabies, ringworm, salmonellosis, tuberculosis, and vesicular stomatitis are of serious public health significance. They cause severe economic losses in animal industries [128]. Rotavirus group A is one of the most common causes of newborn calf diarrhea. In 2013, a group of rotaviruses was discovered in an epizootic outbreak of diarrhea in adult cows, which coincided with a drop in milk output in Japan [129]. Bovine enterovirus is another virus that causes diarrhea in cattle. Abortion, stillbirths, infertility, neonatal mortality, diarrhea, pyrexia, dehydration, and weight loss have all been reported worldwide. NGS technology and quantitative reverse transcription (qRT)-PCR have been used to identify bovine enteroviruses [130].

Pigs are also excellent human disease models and can spread various infections to humans. In addition, pork meat can result in the transmission of different life-threatening conditions. The major zoonotic diseases associated with swine include influenza, ringworm, erysipelas, campylobacteriosis, salmonellosis, cryptosporidiosis, giardiasis, balantidiasis, *E. coli*, brucellosis, and streptococcosis [131]. Pig parasites and their potential to infect humans have lately become a severe public health concern because of recent parasitic disease outbreaks where pigs acted as vectors [132].

Poultry are raised in various cultures, customs, and religious states for food security and nutrition as meat and eggs. Approximately 106 million tons of chicken meat are supplied to the market globally, with a continuous increase compared to beef and pork [133,134]. Zoonotic infections associated with poultry commonly include avian influenza, tuberculosis, erysipelas, ornithosis, cryptococcosis, histoplasmosis, salmonellosis, cryptosporidiosis, campylobacteriosis, and escherichiosis [131]. In March 2004, the chicken genome was the first genome sequenced in any agriculture-related animal species [135].

The zoonotic infections spread by sheep include severe viral diseases that can affect all mammals, such as rabies and other diseases like salmonellosis, listeriosis, Q fever, ringworm, and chlamydiosis [131]. Equine disease models can also be used to study various human diseases. Equine recurrent uveitis is an autoimmune illness that affects horses, yet it is the only valid spontaneous model of human autoimmune uveitis [136]. One of the more prevalent zoonotic parasite diseases is toxoplasmosis. The late 1930s saw the first recognition of *T. gondii*-related disease in humans. The primary mechanism of the vertical transmission of *T. gondii* involves tachyzoites [137]. Although tachyzoites of *T. gondii* have been discovered in the milk of a number of intermediate hosts, including sheep, goats, and cows, a report suggested that acute toxoplasmosis in humans has mostly been associated with the intake of unpasteurized goat’s milk [138,139]. Furthermore, it is considered that *T. gondii* found in livestock meat, is a significant source of infection for people [140].

Many unknown disease-related and zoonosis-causing mutations have been discovered through advances in genome sequencing [141]. The NGS sheds fresh light on the zoonotic spread of microorganisms. High-resolution or ultra-deep sequencing showed the genetic diversity of influenza A and hepatitis E [96,142]. HT-NGS techniques were utilized for the genomic sequencing of influenza (H1N1) from animals. HTS-based metagenomic methods can be utilized to investigate new etiology outbreaks such as understanding host responses to diverse viral infections, gaining information on potential well-known illnesses suspected of having a multi-factorial etiology, and epidemic control through quick diagnosis, high sensitivity, and flexible analysis. Thus, these techniques have the potential to lead to several new advancements in food safety and public health [143].

## 6. The Mechanism of Zoonoses

Bacteria, viruses, parasites, and fungi are the primary pathogens that cause zoonotic diseases [91]. Anti-microbial resistance is a severe global issue affecting both humans and agricultural animals [144]. Adaptive resistance is the product of bacterial survival mechanisms in response to altered environmental conditions; it is attained by horizontal and vertical gene transfer. Viruses rarely encounter optimal environments, and natural selection through mutations enables their survival in extreme conditions [145]. Many mutations in the host may be eradicated through purifying selection [146]. The purifying selection represents the most predominant form of choice as it persistently wipes out newly appearing deleterious mutations in coding regions produced in virus replication [147]. Based on genome composition and host cellular organization, viruses are expected to encounter widely altered selection enforcement, particularly in an advanced organism such as a vertebrate that contains a unique mechanism of immunity, which is the highly specific detection of foreign proteins by certain recognition receptors [148]. An intense mutation rate can cause the production and accumulation of deleterious mutations; however, those deleterious mutations can be eradicated by purifying selection [147]. Austin L. Hughes et.al author indicates that purifying selection is ongoing in nonsynonymous sites and not in synonymous sites, and that there is a more effective action of purifying selection in RNA viruses than in DNA viruses [148]. Moreover, the author suggested that purifying selection is relaxed on exposed proteins of RNA and DNA viruses which are infecting vertebrates, except in the case of those with arthropod vectors (the influence of purifying selection is varied on infection from different hosts) [148]. Arcangeli et al. confirmed the presence of purifying selection in their studies, revealing that ss-RNA-strand small-ruminant lentiviruses (SRLVs) exhibit a high mutation rate and frequent recombination events, but the obtained value of the non-synonymous (dN) and synonymous (dS) substitution (dN/dS) ratio indicated the presence of purifying selection [149]. 

Virus mutation rates vary depending on the polymerase fidelity from high-fidelity DNA polymerases that possess proofreading activity. Mutation in RNA viruses also depends on genome size, with lethal mutations higher in larger RNA genomes. The mutation rate of DNA viruses varies depending on polymerase error, host reaction, and viral error-correction enzymes. Some small DNA viruses do not contain DNA polymerase and use host polymerases for proofreading [150]. Importantly, RNA-dependent RNA and DNA polymerases make more errors, leading to more mutations than DNA polymerases due to a lack of proofreading activity (Figure 3).

Viral mutation rates can also depend on the infected host species [151]; however, the mechanisms associated with virus spillover are still under investigation. Considering the possible impact of spillover events caused by fast mutation and resistance to conventional medications, currently available technological and NGS approaches should be employed to mitigate the effects of such infections on animals.

## 7. HT-NGS and Bioinformatics Simulations for Pathogens Detection

With a predicted global population of approximately 10 billion people by 2050, there will be an unparalleled growth in demand for animal protein, including meat, eggs, milk, and other animal products. The worldwide task will be to provide a food supply that is inexpensive, safe, and sustainable [152,153]. The HT-NGS, paired with computer modeling and algorithm, allows us to effectively diagnose infection in domestic animals and identify known or unknown pathogens [154]. Sequencing technologies enable the screening of vast populations of domesticated animals for genetic variations that mirror human genetic illnesses and allow the development of models that represent uncommon human disorders more precisely. These technologies can facilitate the real-time identification and quantification of aerobic and anaerobic bacteria and fungi.

The genomics revolution provides enormous promise for generating novel insights and disease control techniques as the pathogens of tickborne livestock diseases have been sequenced. Additionally, with the increasing accessibility of genetic resources, the interconnections between species participating in the tick–host–pathogen system can be investigated [155]. In Australasia and Asia, tickborne illnesses have significant adverse economic impacts on cattle operations. Oriental theileriosis is a tickborne illness that affects cattle and is caused by the members of the *Theileria Orientalis* complex. Five genotypes of the *T. Orientalis* complex in 13 cattle samples have been identified using NGS [156]. The viral metagenomics analyses can be used to detect groups of rotaviruses from fecal samples, allowing impartial and thorough diagnoses of diseases in the animal.

Over the last decade, the control of parasitic sheep illnesses has been challenging despite several changes and developments in management, which challenges the safe rearing of sheep in many parts of the world and increases human zoonotic hazards [157]. As large-animal models for biomedical research, sheep are more promising than mice because they have more physiological similarities to humans [157]. Small ruminant lentiviruses (SRLVs) have at least four highly diverse viral genotypes, which persist in the sheep spleen. Whole-genome characterization of SRLV is now possible through NGS [158]. The establishment of genome sequence databases can facilitate prompt and accurate recognition of emerging unknown infections or disease strains, supporting endeavors for curbing widespread contemporary diseases like the coronavirus pandemic.

## 8. Conclusions

Although HT-NGS technology changed sequencing by providing unprecedented depth and accuracy, it still has significant limitations. The generation of short readings is a severe challenge. The so-called “short-read sequencing” that defines all NGS technologies necessitates the use of specialized bioinformatics tools and complex post-processing pipelines, making high-throughput data handling more challenging and increasing the average duration of the analysis. Short-read approaches are often characterized by the use of large equipment and time-consuming experimental processes, as well as substantial bioinformatics analysis. These characteristics of NGS approaches make the testing procedure complicated for post-processing analysis. Studies on variation analysis claim that long-reads have enabled researchers to more easily characterize large insertions, deletions, translocations, and other structural alterations that could be present across the genomes. Longer read lengths contribute to more figurative chromosomal elements, resulting in more contiguous genome reconstructions.

Furthermore, the metagenomic approach in environmental samples tends to multiply mistakes, thus confounding the conclusions concerning pathogen diversity. It is challenging to determine the pathogen virulence in humans or their domestic animals because infections and parasites are so varied. The databases described here can assist us in making predictions and directing the available research to validate the predictions made using bioinformatic databases developed using NGS technology. Regardless of rates or timeframes, the most critical purpose of animal genome research is to enhance our understanding of different breeds’ genome information and control and prevent animal disease spread/diffusion to avert agroeconomic losses and prevent the outbreak of new pandemics.

## Figures and Tables

**Figure 1 life-12-01893-f001:**
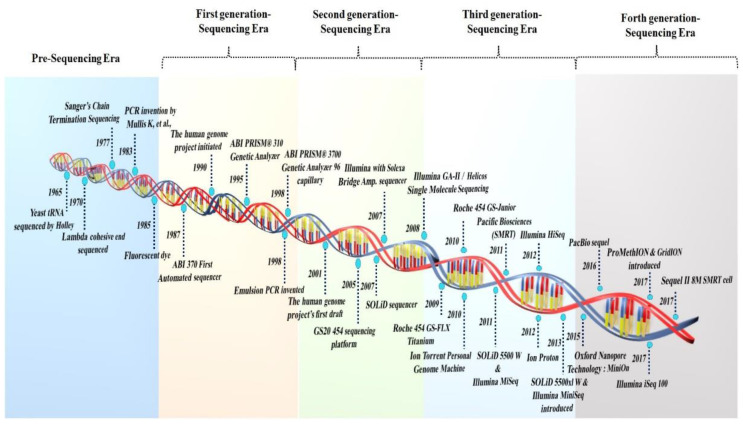
A symmetrical illustration of the period and the development of sequencing platforms.

**Figure 2 life-12-01893-f002:**
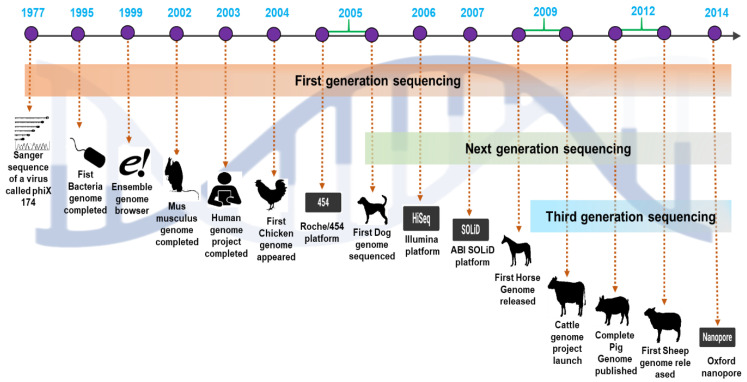
A comprehensive overview of the emergence of next-generation sequencing and the timeline of the valuable domestic animal genome findings.

**Figure 3 life-12-01893-f003:**
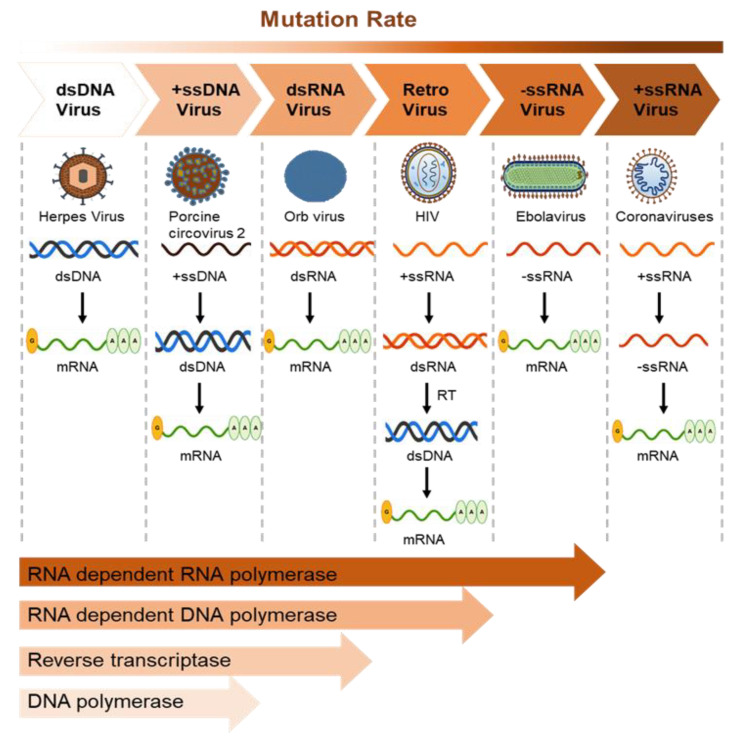
Mutation rates of viruses represent the rate of virus evolution. The average mutation rate of the dsDNA viruses is the lowest owing to the proofreading activity of the harbored DNA polymerase (1.38 × 10^−7^ nucleotide substitutions/site/year in Herpes virus). The ssRNA viruses show the highest mutation rates (2.0 × 10^−3^ nucleotide substitutions/site/year in the Ebola virus, 0.8–2.38 × 10^−3^ in coronaviruses, and 1.21 × 10^−2^ in norovirus). The RNA-dependent RNA and DNA polymerases are more likely to cause mutation than DNA-dependent DNA polymerases as they lack proofreading activity. The mutation rate of reverse transcriptase is higher than that of DNA polymerase; however, RNA viruses show more mutations than retroviruses.

**Table 1 life-12-01893-t001:** High-throughput next-generation sequencing reported in the livestock Sequence Read Archive (SRA) Experiments (https://www.ncbi.nlm.nih.gov/Taxonomy/Browser/wwwtax.cgi? (accessed on 2 November 2022)).

Species	SRA Experiments (Direct Link)
*Bos taurus* (cattle)	65,068
*Sus scrofa domesticus* (domestic swine)	4449
*Ovis aries* (sheep)	13,227
*Equus caballus* (horse)	13,398
*Gallus gallus domesticus* (domestic chicken)	38,902
*Anas platyrhynchos* (mallard)	6649

**Table 2 life-12-01893-t002:** Existing online databases for domestic animal genome-based research (accessed on 2 November 2022).

Animal	Category	Resource	URL	Description	References
Bovine	Cattle Quantitative Trait Locus (QTL) Database	Cattle QTLdb	https://www.animalgenome.org/cgi-bin/QTLdb/BT/index (accessed on 2 November 2022)	The cattle QTL (contains 193,216 QTLs/associations) association data curated from published data (1111 publications). Those QTLs/associations represent 684 different traits.	[70]
	Genome sequence	Bovine Genome Database (BGD)	http://bovinegenome.org/ (accessed on 2 November 2022)	Sequencing of the cattle genome first began in December 2003. The most recent version of BovineMine (BovineMine v1.6) now includes both the ARS-UCD1.2 and UMD3.1 genome assembly, whereas the previous version (BovineMine v1.4) only had UMD3.1.1. JBrowse is compatible with both ARS-UCD1.2 and UMD3.1.	[71]
	Variation database	BGVD	https://animal.nwsuaf.edu.cn/code/index.php/BosVar/ (accessed on 2 November 2022) http://222.90.83.22:88/code/index.php/BosVar (accessed on 2 November 2022)	The BGVD includes information on genomic variants of 432 samples from contemporary cattle worldwide, including ~60.44 million single-nucleotide polymorphisms (SNPs), ~6.86 million indels, and ~76,634 copy number variations with signs of selective sweeps. It can provide information about the selection scores for eight groups of European taurine, Eurasian taurine, East Asian taurine, Chinese indicine, Indian indicine, Africa taurine, Bos indicus, and Bos taurus by using six statistical terms.	[72]
	Bovine SNP database	SNPchiMp	https://bioinformaticshome.com/tools/descriptions/SNPchiMp_v.3.html (accessed on 2 November 2022)	SNPchiMp is a public MySQL database with a web-based interface officially attributed as an Ensembl web-based server. SNPchiMp v.3 analyzed six livestock species, ranging from one species for goats to more than ten for cattle, with a total of 23 SNP arrays. The interface includes SNP mapping information from the most recent genome assembly, information extraction from dbSNP for SNPs detected in all commercially available bovine chips, and identification of SNPs shared by two or more bovine chips.	[73]
	Metabolome database	The Bovine Metabolome	https://bovinedb.ca/ (accessed on 2 November 2022)	It is a free online resource that contains thorough information about small molecule metabolites identified in bovines. It is meant to be used to learn more about bovine biology and the micronutrients contained in bovine tissues and biofluids, as well as to improve beef and dairy cow veterinarian treatment. Serum, ruminal fluid, liver, longissimus thoracis (LT) muscle, semimembranosus (SM) muscle, and testis tissues are all characterized quantitatively in BMDB. Many data fields are connected to various databases (HMDB, PubChem, MetaCyc, ChEBI, UniProt, and GenBank) and applets for visualizing structure and pathways.	[74]
	Proteome database	BoMiProt	http://bomiprot.org/ (accessed on 2 November 2022)	BoMiProt, an online library of bovine milk proteome, contains approximately 3100 proteins from whey, fat globule membranes, and exosomes. Each entry in the database is thoroughly cross-referenced, comprising 397 proteins from various publications with well-defined information on protein function, biochemical characteristics, post-translational modifications, and relevance in milk.	[75]
Sheep	Variantion database	SheepVar	http://222.90.83.22:88/code/index.php/SheepVar (accessed on 2 November 2022)	The database is an online resource led by Yu Jiang (Northwest A&F University, Yangling, Shaanxi, China). This comprehensive SheepVar database includes ~83 M SNPs and ~7 M Indels derived from 1116 samples of seven wild sheep relatives and 135 domestic sheep breeds. This database was curated by analyzing 64 wild sheep samples and 1052 domestic sheep samples and also provides two ways to view SNPs and indels, one is interactive tables and geographical maps, and the other is in Gbrowse format.	
	Quantitative Trait Locus (QTL) Database	Sheep QTLdb	https://www.animalgenome.org/cgi-bin/QTLdb/OA/index (accessed on 2 November 2022)	The Sheep QTLdb is valuable for population genetic research. The frequency of these tools used for searching by chromosomes, traits, breeds, publications, and candidate genes. Sheep QTLdb contains 4416 QTLs/associations from 226 publications. Those QTLs/associations represent 266 different traits.	[76]
	International Sheep Genomics Consortium	ISheep	https://www.sheephapmap.org/ (accessed on 2 November 2022)	The ISGC helps researchers identify genetic areas and genes that influence sheep characteristics. This database serves as a backbone for ruminant species when coupled with data from other ruminant genome sequences. The database contains sheep genome assemblies and variants of 935 sheep representing 69 breeds from 21 countries. In addition to providing a genetic resource for animal biomedical research models, this assembly is a genomic resource for humans.	[77]
	Sheep Genomes Database	Sheep Genomes DB	https://sheepgenomesdb.org/ (accessed on 2 November 2022)	The USDA AFRI-funded Sheep Genomes Database is a project of the International Sheep Genomics Consortium that builds on the consortium’s recent achievement of creating and sharing the Oar rambouillet v1.0 genome. It gathers and facilitates sheep genomic data, detects variants, and downloads SNP and CNV data from sheep genomes.	
Pig	Pig Pan-genome Database	PIGPAN	http://222.90.83.22:88/code/index.php/panPig (accessed on 2 November 2022)	Third-generation sequencing technology was used to assemble the 2.4 Gb Duroc genome (Sscrofa11.1) and 72.5 Mb pan-sequences from 11 significant local European and Chinese pig varieties. The pan-genome offers a rich data set for the scientific community, which would support the pig genome’s development.	[78]
	Swine genome sequencing data	SGSC	https://www.igb.illinois.edu/labs/schook/sgsc/index.php (accessed on 2 November 2022)	It was established in September 2003 to promote biomedical research for animal health. It supports creating DNA-based technologies and products from swine genome sequencing data.	[79]
	Pig Expression Data Explorer	PEDE	https://pede.dna.affrc.go.jp/ (accessed on 2 November 2022)	The Animal Genome Research Program in Japan, which is operated by the JATAFF-Institute and National Institute of Agrobiological Sciences, maintains this database website. In conjunction with the NIAS DNA bank, the Animal Genome Database, the SNP Linkage map, and the RH map are resources that include PEDE. Pig cSNPs (SNPs in cDNA) were found using the PolyPhred program on the PEDE EST assembly.	[80,81]
	Pig Quantitative Trait Locus (QTL) Database	Pig QTLdb	https://www.animalgenome.org/cgi-bin/QTLdb/SS/index (accessed on 2 November 2022)	The Pig QTL (35,846 QTLs/associations) association data is curated from published data (773 publications). Those QTLs/associations represent 693 different traits.	[70]
Chicken and duck	Chicken SNP Database	ChickenSD	https://ngdc.cncb.ac.cn/chickensd/ (accessed on 2 November 2022)	A total of 865 samples were used to identify approximately 33 million whole genome non-redundant SNPs in ChickenSD (167 wild, 697 domesticated, and 1 hybrids). A total of 865 samples were used to identify approximately 33 million whole genome non-redundant SNPs in ChickenSD (167 wild, 697 domesticated, and 1 hybrid). The Chinese Academy of Sciences BIG Data Center, Beijing Institute of Genomics (BIG), is in charge of creating and maintaining this database (CAS). The Kunming Institute of Zoology (KIZ), part of the Chinese Academy of Sciences, was tasked with gathering and curating the data (CAS).	[82,83]
	Chicken Quantitative Trait Locus (QTL) Database	Chicken QTLdb	https://www.animalgenome.org/cgi-bin/QTLdb/GG/index (accessed on 2 November 2022)	The Pig QTL (16,656 QTLs/associations) association data is curated from published data (376 publications). Those QTLs/associations represent 370 different traits.	[70]
	Gene expression	GEISHA	http://geisha.arizona.edu (accessed on 2 November 2022)	GEISHA is a chicken embryo in situ hybridization gene expression database and genomics resource. More than 36,000 pictures of whole-embryo in situ hybridizations and embryo portions from embryonic days 0–5, as well as some older embryo data focusing on late-developing tissues, are currently available in the GEISHA database.	[84]
Horse	Genetic variation annotation	EquCab2.0 and 3.0	https://www.ncbi.nlm.nih.gov/assembly/GCF_000002305.2/ (accessed on 2 November 2022) https://www.ncbi.nlm.nih.gov/assembly/GCF_002863925.1/ (accessed on 2 November 2022)	EquCab2.0 is a publicly available genetic variation annotation reference genome assembly for the domesticated horse, assembled in 2007. EquCab3.0 is the updated reference genome assembly. EquCab2.0 was compiled by sequencing the whole genomes of six horses from six different breeds. One thousand three hundred million reads with coverage between 15× to 24× were generated for these six horse breeds. After rigorous filtration, 17,514,723 SNPs and 1923,693 indels, as well as an average of 1540 CNVs and 3321 structural variations per horse, were identified and functionally annotated.	[85,86]
	Methylated regions	HEpd	http://www.primate.or.kr/hepd (accessed on 2 November 2022)	The HEpd database contains information on differentially methylated regions and epigenetic changes between two horse subspecies. It employs a gene index to compare the methylation status in a gene area. Users can filter highly methylated sites beyond a user-defined threshold using this database.	[87,88]
Common database	Animal metagenomes	AnimalMetagenome DB	https://github.com/boyNextDooooor/AnimalMetagenomeDB (accessed on 2 November 2022) https://doi.org/10.6084/m9.figshare.19728619 (accessed on 2 November 2022)	AnimalMetagenomeDB combines metagenomic sequencing data with host information to help users discover relevant data. Animal metagenomic data may be seen, searched for, and downloaded by users. Metadata for 82,097 metagenomes from four domestic animals (bovines, sheep, horses, and pigs) and 540 wild species are included in the AnimalMetagenome DB version 1.0. These metagenomes span 15 years of research, 73 nations, 1044 investigations, 63,214 amplicon sequencing data points, and 10,672 whole genome sequencing data points.	[68]

## Data Availability

Not applicable.

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
