# Peer review of "The Innovative Informatics Approaches of High-Throughput Technologies in Livestock: Spearheading the Sustainability and Resiliency of Agrigenomics Research"

_life, 2022, doi:10.3390/life12111893_

Round 1

Reviewer 1 Report

This review introduced the application of high-throughput technologies approaches in livestock. The author provides relevant information and resources for readers to quickly understand these. But the quality of references cited needs to be improved. The content of this review is too broad, and most are brief introductions without details. I suggest that the authors should check whether the text, figures and tables are consistent with the title and subtitles.

Line 84-88, the abbreviations of HGS, Tn-seq, TraDIS, INseq, and HITS should be explained when they first appear.

Line 99-133, That's the point. You wrote it too simply. Please introduce the characteristics of each generation sequencing technology in detail, especially the third generation sequencing technology. The quality of references cited needs to be improved. There are many top journal papers on the latest sequencing technologies that were not cited.

Line 151-152, the abbreviation SRA need annotation. Please check whether the website provided here is correct.

Table 1, Please add the English name of the species in addition to the Latin names.

Table 2, please check whether all these websites are correct and can be opened by the readers. For example, I found that the website of PiGenome (http://pigenome.nabc.go.kr/) can not be connected.

Table 3, please check whether all these websites are correct and can be opened by the readers. For example, I found that the website of MetaGeneMark for MetaGenome Gene Finding can not be opened.

Line 363-368, bacterial antibiotic resistance has little to do with the mechanism of zoonoses.

Figure 2. Overview of bacterial antibiotic resistance mechanisms. This figure has little to do with the mechanism of zoonoses, it is recommended to delete this figure.

Line 396, An incomplete sentence or paragraph.

Author Response

Comments and Suggestions for Authors

 Reviewer #1

This review introduced the application of high-throughput technologies approaches in livestock. The author provides relevant information and resources for readers to quickly understand these. But the quality of references cited needs to be improved. The content of this review is too broad, and most are brief introductions without details. I suggest that the authors should check whether the text, figures and tables are consistent with the title and subtitles.

[Response] First and foremost, we express my gratitude for informing us about the inaccuracy in the manuscript's title. I'd want to thank you for notifying us of the inadvertent error that has been scribbled in this article.

 Comment-1: Line 84-88, the abbreviations of HGS, Tn-seq, TraDIS, INseq, and HITS should be explained when they first appear.

[Response] We addressed the mistake and incorporated it into the document (lines 91~97).

Comment-2: Line 99-133, That's the point. You wrote it too simply. Please introduce the characteristics of each generation sequencing technology in detail, especially the third generation sequencing technology. The quality of references cited needs to be improved. There are many top journal papers on the latest sequencing technologies that were not cited.

[Response] We addressed the reviewers' concerns and incorporated the descriptions into the manuscript. We revise and rewrite section 2 “Era of Sequencing Technologies Development” to improve readability and comprehension (lines 115~247).

[2. Era of Sequencing Technologies Development

The period of current sequencing technology began in 1959, with the discovery of the bacteriophage ϕX174 genome, when Sinsheimer purified the first DNA molecule to homogeneity [11]. However, the discovery of type II restriction enzymes by Hamilton Smith and colleagues (in 1970-1972) substantially changed the approach to modern genetics, and DNA sequencing could not have occurred without this discovery [12] (Figure 1). The first whole genome of a bacteriophage (MS2) was announced, and in 1972 Sanger's "plus and minus" sequencing method was identified as a crucial transition technology leading to modern methods [13]. Thus, current DNA sequencing began in 1977 with the invention of the chemical approach of Maxam and Gilbert and the dideoxy method of Sanger, Nicklen, and Coulson [14].

2.1. The First Generation of Sequencing Technologies

The throughput sequencing methods can be divided into three generations of sequencing technologies: post-monopoly era sequencing platforms, second-generation, and third- and fourth-generation sequencing platforms [15]. In 1986, near the end of the capillary electrophoresis technique sequencing era, further, development resulted in an automated fluorescent technique to sequence a genome region. It can be considered the primary technology in the "first generation" of automated DNA sequencing, which was reported using Applied Biosystems (ABI) fluorescent sequencing [16]. Efforts are being made to improve sequencing techniques, allowing for the development of increasingly automated DNA sequencing equipment with fluorometric detection and enhanced sensing employing capillary-based electrophoresis. The Welcome Trust and Medical Research Council integrated a global public effort to sequence the Human Genome Project. The project was initially begun in 1990 and completed in 2003. Although the Sanger technique was used to sequence the first 3.0 billion bp of the human genome (released in 2000), the human reference genome has only covered the euchromatic part of the genome, rendering crucial heterochromatic regions incomplete. The Genome Reference Consortium (GRC) released the current human reference genome in 2013 and most recently updated it in 2019 (GRCh38.p13). This reference has evolved over the past 20 years and can be attributed to the Human Genome Project [17].

Several unique sequencing technologies have been developed two decades after the advent of electrophoretic techniques for DNA sequencing. The words "next-generation" and/or "massively parallel" DNA sequencing is used to refer to DNA HTS technologies that can sequence a large number of distinct DNA sequences in a single reaction. Sanger-based "topdown" techniques need to characterization of large clones by low-resolution mapping in microtiter plate wells, whereas massively parallel approaches do not. The main premise of NGS approaches based on DNA ligase covalently attaching synthetic DNA adapters to each of the targeted fragment ends and in situ amplification on a solid surface. Solexa technology was developed in 1998, and 454 Life Science in 2000. However, the GS20 454 sequencing platform debuted in 2005 and was the first non-Sanger-based commercialized technology [18]. Roche 454, the first commercial NGS platform, employed large-scale parallel pyrosequencing chemistry to identify base pair sequences with high throughput and lower sequencing costs per base than Sanger sequencing [19].

2.2. The Second Generation of Sequencing Technologies

The NGS methodology has been used in many fields, including transcriptome analysis, de novo assembly, genotyping, targeted and whole genome sequencing, and the detection of SNPs, copy number variation, exome, protein-protein interactions, and genome methylation. The Roche 454 genome technology, which is based on Melamede's sequencing-by-synthesis (SBS) theory (1985), was the first next-generation system to be commercially viable and uses pyrophosphate to identify pyrophosphate generated during DNA synthesis. The Roche 454 GS system was initially released in 2005, and in 2008 it was updated to the Roche GS-FLX 454 Titanium system. The GS run processor and additional work in 2009 streamlined library preparation and data processing. Roche employed a GS FLX+ sequencer capable of reading 400-600 million base pairs each run with maximum pair read lengths of 1,000, however, Roche-454 was phased out in 2016 [20].

The SOLiD platform, developed by Harvard Medical School and the Howard Hughes Medical Institute, was commercially released by ABI in October 2007 and generated 4 Gb of sequencing data within the six days run[12]. This sequencing system employs the sequencing-by-ligation method of oligonucleotide ligation and detection. The SOLiD sequence is based on color coding, which is decoded to produce the basic sequence. However, incorrect color coding might result in decoding errors. Balasubramanian and Klenerman utilized fluorescently tagged nucleotides in the middle of the 1990s to observe a single polymerase molecule migrate[12]. In June 2006, the first Solexa sequencer was launched, and Illumina entered the industry in 2006, bought Solexa in 2007, and gradually progressed the NGS industry[16]. A paired-end module for the sequencer with new optics and camera components was included in the Genome Analyzer II in 2008 as a result of further advancements in the Illumina method[12].

Ion Torrent Systems Inc., in 2010, invented the first commercial sequencing method that did not rely on dye-labeled oligonucleotides and expensive optics. It monitors H ions generated during base incorporation and is specifically suited for amplicon sequencing. Despite its benefits, Ion Torrent's read accuracy remains a major challenge. The high rate of mistakes induced by the noisy sequencer signal is translated into a nucleotide sequence. Furthermore, the signal decays over time, resulting in a drop in the signal-to-noise ratio [21]. Polonator, a polony sequencing machine, was invented by Dr. George Church's group at Harvard Medical School in 2009. Polony sequencing, a non-electrophoretic sequencing technology, can read millions of immobilized DNA sequences simultaneously at a lower cost per nucleotide than conventional Sanger sequencing. The fundamental limitation of this method is non-uniform amplification, which results in decreased sequencing accuracy and a read length of just 26 bp [22]. The second-generation 454 GS-FLX, Illumina, and SOLiD sequencing systems are not sensitive enough to detect individual single-molecule template extensions, whereas "third- or fourth-generation sequencers" are "single molecular"-type sequencers, such as the Heliscope, PacBio, and Oxford nanopore sequencers, which do not require pre-amplification steps and are more sensitive and precise.

2.3. Third-generation sequencing platforms

Third-generation sequencing systems do not include an amplification step during library creation. They allow single-molecule sequencing with average read lengths reaching 6-8 kbp and maximum read lengths exceeding 30-150 kbp. Based on real-time imaging, the SMRT sequencing technology parallelizes data from a DNA polymerase and conducts uninterrupted template-directed synthesis. By emphasizing length, it breaks out of existing short-read HTS instruments. In 2011, Pacific Biosciences made SMRT sequencing commercially available [23]. PacBio technology yields read durations ranging from 1000 to 3000 bp on average [12].

Illumina technology employs DNA colony sequencing, which is based on reversible dye terminator sequencing via synthesis chemistry. The Illumina sequencing-by-synthesis method is the most extensively used NGS technology because it provides precise read alignment and improved indel identification [19]. Early in 2010, Illumina introduced HiSeq 2000, and continued research on cutting-edge flow cells for Illumina HiSeq technology led to the numerous novel sequencing platforms introduced from 2011 to 2018. Illumina has produced popular sequencing systems, including MiSeq, HiSeq, and NovaSeq [24]. Current NGS methods are at least 100 times quicker than traditional Sanger sequencing. Using NGS, complete genome sequences may be retrieved, providing fast and comprehensive information [25]. As a result, NGS technology is frequently employed to monitor gene expression across an organism's genome.

Further development of HT next-generation platforms such as the GeneReader NGS technique, The 10X Genomics platform, The SeqStudioTM Genetic analyzer, the Bionano SaphyTM genomics platform, the fluorescence resonance energy transfer based GnuBio platform (Bio-Rad, Hercules, CA), GenapSys, NanoString Technologies, an electron microscopy-based Electron Optica system and Firefly (Illumina, San Diego, CA), , nanopore sequencing by Genia (Roche, Basel, Switzerland),can revolutionize biological science through the ability to sequence more samples at higher depths, producing more insightful data in less time and at a lower cost per sample [26].

2.4. Fourth-generation sequencing platforms

Following the three generations, a new type of sequencer was recently developed, represented by the PacBio sequencer and Nanopore sequencer, known as fourth- generation sequencing [27]. Oxford Nanopore Technologies (ONT) introduced two new TGS systems, MinION, PromethION, and GridION, in 2012, enabling the direct electronic study of DNA, RNA, proteins, and single molecules. This method uses nanopores and an exonuclease-based "deconstruction sequencing" approach. In 2014-2015, the MinIONs were distributed to selected laboratories for beta testing. Nanopore technology can provide real-time sequencing of single molecules for as little as $25-$40 per Gb of sequence data. Because alignment and assembly are more straightforward using nanopore technology, data processing is simpler than short-read sequencers. GridION tested up to five MinION Flow Cells simultaneously; it is a simplified benchtop infrastructure. It is ideal for labs with various applications that require the benefits of nanopore sequencing, such as facile library preparation, real-time analysis, and lengthy reads. PromethION is meant for HT and employs the same chemistry as MinION and GridION, which are intended for realtime usage. However, based on the number of samples, it has a high fidelity for DNA and RNA sequencing. It is a rapid sequencing method, and nanopore technology may represent the future of sequencing.]

References

  1. Sinsheimer, R.L. A single-stranded DNA from bacteriophage phi X174. Brookhaven Symp Biol 1959, No 12, 27-34.
  2. Ghosh, M.; Sharma, N.; Singh, A.K.; Gera, M.; Pulicherla, K.K.; Jeong, D.K. Transformation of animal genomics by next-generation sequencing technologies: a decade of challenges and their impact on genetic architecture. Crit Rev Biotechnol 2018, 38, 1157-1175, doi:10.1080/07388551.2018.1451819.
  3. Hutchison, C.A., 3rd. DNA sequencing: bench to bedside and beyond. Nucleic acids research 2007, 35, 6227-6237, doi:10.1093/nar/gkm688.
  4. Sanger, F.; Nicklen, S.; Coulson, A.R. DNA sequencing with chain-terminating inhibitors. Proceedings of the National Academy of Sciences of the United States of America 1977, 74, 5463-5467, doi:10.1073/pnas.74.12.5463.
  5. Pei, S.; Liu, T.; Ren, X.; Li, W.; Chen, C.; Xie, Z. Benchmarking variant callers in next-generation and third-generation sequencing analysis. Brief Bioinform 2021, 22, bbaa148, doi:10.1093/bib/bbaa148.
  6. Liu, L.; Li, Y.; Li, S.; Hu, N.; He, Y.; Pong, R.; Lin, D.; Lu, L.; Law, M. Comparison of next-generation sequencing systems. J Biomed Biotechnol 2012, 2012, 251364, doi:10.1155/2012/251364.
  7. Nurk, S.; Koren, S.; Rhie, A.; Rautiainen, M.; Bzikadze, A.V.; Mikheenko, A.; Vollger, M.R.; Altemose, N.; Uralsky, L.; Gershman, A.; et al. The complete sequence of a human genome. Science (New York, N.Y.) 2022, 376, 44-53, doi:10.1126/science.abj6987.
  8. Diaz, M.H.; Winchell, J.M. The evolution of advanced molecular diagnostics for the detection and characterization of Mycoplasma pneumoniae. Frontiers in microbiology 2016, 7, 232.
  9. Voelkerding, K.V.; Dames, S.A.; Durtschi, J.D. Next-generation sequencing: from basic research to diagnostics. Clinical chemistry 2009, 55, 641-658.
  10. Microbiology, A.A.o. Applications of Clinical Microbial Next‐Generation Sequencing: Report on an American Academy of Microbiology colloquium held in Washington, DC, in April 2015. 2016.
  11. Golan, D.; Medvedev, P. Using state machines to model the Ion Torrent sequencing process and to improve read error rates. Bioinformatics 2013, 29, i344-351, doi:10.1093/bioinformatics/btt212.
  12. Shendure, J.; Porreca, G.J.; Reppas, N.B.; Lin, X.; McCutcheon, J.P.; Rosenbaum, A.M.; Wang, M.D.; Zhang, K.; Mitra, R.D.; Church, G.M. Accurate multiplex polony sequencing of an evolved bacterial genome. Science (New York, N.Y.) 2005, 309, 1728-1732, doi:10.1126/science.1117389.
  13. Barba, E.; Tsermpini, E.-E.; Patrinos, G.P.; Koromina, M. Genome Informatics Pipelines and Genome Browsers. In Applied Genomics and Public Health; Elsevier: 2020; pp. 149-169.
  14. Gourle, H.; Karlsson-Lindsjo, O.; Hayer, J.; Bongcam-Rudloff, E. Simulating Illumina metagenomic data with InSilicoSeq. Bioinformatics 2019, 35, 521-522, doi:10.1093/bioinformatics/bty630.
  15. Perez-Enciso, M.; Ferretti, L. Massive parallel sequencing in animal genetics: wherefroms and wheretos. Anim Genet 2010, 41, 561-569, doi:10.1111/j.1365-2052.2010.02057.x.
  16. Goodwin, S.; McPherson, J.D.; McCombie, W.R. Coming of age: ten years of next-generation sequencing technologies. Nat Rev Genet 2016, 17, 333-351, doi:10.1038/nrg.2016.49.
  17. Feng, Y.; Zhang, Y.; Ying, C.; Wang, D.; Du, C. Nanopore-based fourth-generation DNA sequencing technology. Genomics Proteomics Bioinformatics 2015, 13, 4-16, doi:10.1016/j.gpb.2015.01.009.

Comment-3: Line 151-152, the abbreviation SRA need annotation. Please check whether the website provided here is correct.

 [Response] We addressed the reviewers' concerns (lines 274-277).

Comment-4: Table 1, Please add the English name of the species in addition to the Latin names.

[Response] The correction has been included in Table 1.

Comment-5: Table 2, please check whether all these websites are correct and can be opened by the readers. For example, I found that the website of PiGenome (http://pigenome.nabc.go.kr/) can not be connected.

 [Response] We rectified the web page validity and integrated new websites into Table 2.

 Comment-6: Table 3, please check whether all these websites are correct and can be opened by the readers. For example, I found that the website of MetaGeneMark for MetaGenome Gene Finding can not be opened.

[Response] We rectified the web page validity and integrated new websites and the description column into Table 3.

 Comment-7: Line 363-368, bacterial antibiotic resistance has little to do with the mechanism of zoonoses.

[Response] We addressed the reviewers' concerns (lines 534~538; 548~553).

[Arcangeli et al. confirmed the presence of purifying selection in their studies, revealing that ss-RNA strand small ruminant lentiviruses (SRLVs) exhibit a high mutation rate and frequent recombination events, but the obtained value of non-synonymous (dN) and synonymous (dS) substitution (dN/dS) ratio indicated the presence of purifying selection [150].

[Viral mutation rates can also depend on the infected host species [152]; however, the mechanisms associated with virus spillover are still under investigation. Considering the possible impact of spillover events caused by fast mutation and resistance to conventional medications, currently available technological and NGS approaches should be employed to mitigate the effects of such infection on animals.]

References

  1. Arcangeli, C.; Torricelli, M.; Sebastiani, C.; Lucarelli, D.; Ciullo, M.; Passamonti, F.; Giammarioli, M.; Biagetti, M. Genetic Characterization of Small Ruminant Lentiviruses (SRLVs) Circulating in Naturally Infected Sheep in Central Italy. Viruses 2022, 14, 686.
  2. Combe, M.; Sanjuan, R. Variability in the mutation rates of RNA viruses. Future Virology 2014, 9, 605-615, doi:10.2217/Fvl.14.41.

Comment-8: Figure 2. Overview of bacterial antibiotic resistance mechanisms. This figure has little to do with the mechanism of zoonoses, it is recommended to delete this figure.

[Response] The figure has been removed from the manuscript.

Comment-9: Line 396, An incomplete sentence or paragraph.

[Response] The sentence is a subtitle. We corrected it (line 564).

Reviewer 2 Report

Dear Authors,

this review is very valid because very interesting and useful for researcher that dealing with the impactful NGS topic and its application in Animal disease control (including zoonoses) for agrogenomics improvement (like in my case).

The manuscript is well written and well organized, allowing a clear reading and a full comprehension. I have only some suggestions and little comments that are the following.

Please change format, justifying the text. 

The are some orthographic and spelling mistakes, some italic forms and some acronymous not defined. 

Please control pages numeration.

I suggest to elucidate differences with Sanger sequencing approach, highlighting weakness and strenght aspects, reporting some recent references about its application in animal infection monitoring or SNPs detection in genetic resistance/susceptibility to animal diseases.

Line 243: "desiderable genetic traits" is better than "wanted genes"

3.5 section: probably you wanted to write "insight".

Lines 285-297: please cite at least a reference about the statement reported in the text.

Line 353-361: references are lacking. Furthermore, please implement this paragraph. Also T. gondii in an important neglected but spread and ri-emerged zoonoses, so I suggest to include it.

Phenomenon of purifying selection has to be more detailed. Some references touch and deal with this argue (such as Arcangeli et al 2022.)

I suggest to mention Covid/Sars Cov2 in pandemic era when you referred to spillover question.

Line 396: the sentence is incomplete 

Line 397: please, reformulate the misleading sentence in "the considering/the consideration of the growing global demand..." 

Line 456: I'd specify "control and prevent animal disease spread/diffusion".

Please check the format of reference list in line of "Life" Journal required format.

Author Response

Reviewer #2

this review is very valid because very interesting and useful for researcher that dealing with the impactful NGS topic and its application in Animal disease control (including zoonoses) for agrogenomics improvement (like in my case).

The manuscript is well written and well organized, allowing a clear reading and a full comprehension. I have only some suggestions and little comments that are the following.

[Response] We appreciate the reviewer for supporting us in the further improvement of this manuscript. We carefully considered the comments and tried our best to address every one of them. 

 Comment-1: Please change format, justifying the text. 

[Response] We used a formatted word file provided by Journal.

Comment-2: The are some orthographic and spelling mistakes, some italic forms and some acronymous not defined.

 [Response] We rectified the mistake and integrated it into the document. To improve readability and comprehension, we revise and rewrite certain paragraphs in the manuscript.

Comment-3: Please control pages numeration.

 [Response] We used Journal formatted word file and confirmed all page numeration.

Comment-4: I suggest to elucidate differences with Sanger sequencing approach, highlighting weakness and strenght aspects, reporting some recent references about its application in animal infection monitoring or SNPs detection in genetic resistance/susceptibility to animal diseases.

 [Response] We addressed the reviewers' concerns and incorporated the descriptions into the manuscript. We revise and rewrite section 2 “Era of Sequencing Technologies Development” with improve readability and comprehension (lines 115~247).

[2. Era of Sequencing Technologies Development

The period of current sequencing technology began in 1959, with the discovery of the bacteriophage ϕX174 genome, when Sinsheimer purified the first DNA molecule to homogeneity [11]. However, the discovery of type II restriction enzymes by Hamilton Smith and colleagues (in 1970-1972) substantially changed the approach to modern genetics, and DNA sequencing could not have occurred without this discovery [12] (Figure 1). The first whole genome of a bacteriophage (MS2) was announced, and in 1972 Sanger's "plus and minus" sequencing method was identified as a crucial transition technology leading to modern methods [13]. Thus, current DNA sequencing began in 1977 with the invention of the chemical approach of Maxam and Gilbert and the dideoxy method of Sanger, Nicklen, and Coulson [14].

2.1. The First Generation of Sequencing Technologies

The throughput sequencing methods can be divided into three generations of sequencing technologies: post-monopoly era sequencing platforms, second-generation, and third- and fourth-generation sequencing platforms [15]. In 1986, near the end of the capillary electrophoresis technique sequencing era, further, development resulted in an automated fluorescent technique to sequence a genome region. It can be considered the primary technology in the "first generation" of automated DNA sequencing, which was reported using Applied Biosystems (ABI) fluorescent sequencing [16]. Efforts are being made to improve sequencing techniques, allowing for the development of increasingly automated DNA sequencing equipment with fluorometric detection and enhanced sensing employing capillary-based electrophoresis. The Welcome Trust and Medical Research Council integrated a global public effort to sequence the Human Genome Project. The project was initially begun in 1990 and completed in 2003. Although the Sanger technique was used to sequence the first 3.0 billion bp of the human genome (released in 2000), the human reference genome has only covered the euchromatic part of the genome, rendering crucial heterochromatic regions incomplete. The Genome Reference Consortium (GRC) released the current human reference genome in 2013 and most recently updated it in 2019 (GRCh38.p13). This reference has evolved over the past 20 years and can be attributed to the Human Genome Project [17].

Several unique sequencing technologies have been developed two decades after the advent of electrophoretic techniques for DNA sequencing. The words "next-generation" and/or "massively parallel" DNA sequencing is used to refer to DNA HTS technologies that can sequence a large number of distinct DNA sequences in a single reaction. Sanger-based "topdown" techniques need to characterization of large clones by low-resolution mapping in microtiter plate wells, whereas massively parallel approaches do not. The main premise of NGS approaches based on DNA ligase covalently attaching synthetic DNA adapters to each of the targeted fragment ends and in situ amplification on a solid surface. Solexa technology was developed in 1998, and 454 Life Science in 2000. However, the GS20 454 sequencing platform debuted in 2005 and was the first non-Sanger-based commercialized technology [18]. Roche 454, the first commercial NGS platform, employed large-scale parallel pyrosequencing chemistry to identify base pair sequences with high throughput and lower sequencing costs per base than Sanger sequencing [19].

2.2. The Second Generation of Sequencing Technologies

The NGS methodology has been used in many fields, including transcriptome analysis, de novo assembly, genotyping, targeted and whole genome sequencing, and the detection of SNPs, copy number variation, exome, protein-protein interactions, and genome methylation. The Roche 454 genome technology, which is based on Melamede's sequencing-by-synthesis (SBS) theory (1985), was the first next-generation system to be commercially viable and uses pyrophosphate to identify pyrophosphate generated during DNA synthesis. The Roche 454 GS system was initially released in 2005, and in 2008 it was updated to the Roche GS-FLX 454 Titanium system. The GS run processor and additional work in 2009 streamlined library preparation and data processing. Roche employed a GS FLX+ sequencer capable of reading 400-600 million base pairs each run with maximum pair read lengths of 1,000, however, Roche-454 was phased out in 2016 [20].

The SOLiD platform, developed by Harvard Medical School and the Howard Hughes Medical Institute, was commercially released by ABI in October 2007 and generated 4 Gb of sequencing data within the six days run[12]. This sequencing system employs the sequencing-by-ligation method of oligonucleotide ligation and detection. The SOLiD sequence is based on color coding, which is decoded to produce the basic sequence. However, incorrect color coding might result in decoding errors. Balasubramanian and Klenerman utilized fluorescently tagged nucleotides in the middle of the 1990s to observe a single polymerase molecule migrate[12]. In June 2006, the first Solexa sequencer was launched, and Illumina entered the industry in 2006, bought Solexa in 2007, and gradually progressed the NGS industry[16]. A paired-end module for the sequencer with new optics and camera components was included in the Genome Analyzer II in 2008 as a result of further advancements in the Illumina method[12].

Ion Torrent Systems Inc., in 2010, invented the first commercial sequencing method that did not rely on dye-labeled oligonucleotides and expensive optics. It monitors H ions generated during base incorporation and is specifically suited for amplicon sequencing. Despite its benefits, Ion Torrent's read accuracy remains a major challenge. The high rate of mistakes induced by the noisy sequencer signal is translated into a nucleotide sequence. Furthermore, the signal decays over time, resulting in a drop in the signal-to-noise ratio [21]. Polonator, a polony sequencing machine, was invented by Dr. George Church's group at Harvard Medical School in 2009. Polony sequencing, a non-electrophoretic sequencing technology, can read millions of immobilized DNA sequences simultaneously at a lower cost per nucleotide than conventional Sanger sequencing. The fundamental limitation of this method is non-uniform amplification, which results in decreased sequencing accuracy and a read length of just 26 bp [22]. The second-generation 454 GS-FLX, Illumina, and SOLiD sequencing systems are not sensitive enough to detect individual single-molecule template extensions, whereas "third- or fourth-generation sequencers" are "single molecular"-type sequencers, such as the Heliscope, PacBio, and Oxford nanopore sequencers, which do not require pre-amplification steps and are more sensitive and precise.

2.3. Third-generation sequencing platforms

Third-generation sequencing systems do not include an amplification step during library creation. They allow single-molecule sequencing with average read lengths reaching 6-8 kbp and maximum read lengths exceeding 30-150 kbp. Based on real-time imaging, the SMRT sequencing technology parallelizes data from a DNA polymerase and conducts uninterrupted template-directed synthesis. By emphasizing length, it breaks out of existing short-read HTS instruments. In 2011, Pacific Biosciences made SMRT sequencing commercially available [23]. PacBio technology yields read durations ranging from 1000 to 3000 bp on average [12].

Illumina technology employs DNA colony sequencing, which is based on reversible dye terminator sequencing via synthesis chemistry. The Illumina sequencing-by-synthesis method is the most extensively used NGS technology because it provides precise read alignment and improved indel identification [19]. Early in 2010, Illumina introduced HiSeq 2000, and continued research on cutting-edge flow cells for Illumina HiSeq technology led to the numerous novel sequencing platforms introduced from 2011 to 2018. Illumina has produced popular sequencing systems, including MiSeq, HiSeq, and NovaSeq [24]. Current NGS methods are at least 100 times quicker than traditional Sanger sequencing. Using NGS, complete genome sequences may be retrieved, providing fast and comprehensive information [25]. As a result, NGS technology is frequently employed to monitor gene expression across an organism's genome.

Further development of HT next-generation platforms such as the GeneReader NGS technique, The 10X Genomics platform, The SeqStudioTM Genetic analyzer, the Bionano SaphyTM genomics platform, the fluorescence resonance energy transfer based GnuBio platform (Bio-Rad, Hercules, CA), GenapSys, NanoString Technologies, an electron microscopy-based Electron Optica system and Firefly (Illumina, San Diego, CA), , nanopore sequencing by Genia (Roche, Basel, Switzerland),can revolutionize biological science through the ability to sequence more samples at higher depths, producing more insightful data in less time and at a lower cost per sample [26].

2.4. Fourth-generation sequencing platforms

Following the three generations, a new type of sequencer was recently developed, represented by the PacBio sequencer and Nanopore sequencer, known as fourth- generation sequencing [27]. Oxford Nanopore Technologies (ONT) introduced two new TGS systems, MinION, PromethION, and GridION, in 2012, enabling the direct electronic study of DNA, RNA, proteins, and single molecules. This method uses nanopores and an exonuclease-based "deconstruction sequencing" approach. In 2014-2015, the MinIONs were distributed to selected laboratories for beta testing. Nanopore technology can provide real-time sequencing of single molecules for as little as $25-$40 per Gb of sequence data. Because alignment and assembly are more straightforward using nanopore technology, data processing is simpler than short-read sequencers. GridION tested up to five MinION Flow Cells simultaneously; it is a simplified benchtop infrastructure. It is ideal for labs with various applications that require the benefits of nanopore sequencing, such as facile library preparation, real-time analysis, and lengthy reads. PromethION is meant for HT and employs the same chemistry as MinION and GridION, which are intended for realtime usage. However, based on the number of samples, it has a high fidelity for DNA and RNA sequencing. It is a rapid sequencing method, and nanopore technology may represent the future of sequencing.]

References

  1. Sinsheimer, R.L. A single-stranded DNA from bacteriophage phi X174. Brookhaven Symp Biol 1959, No 12, 27-34.
  2. Ghosh, M.; Sharma, N.; Singh, A.K.; Gera, M.; Pulicherla, K.K.; Jeong, D.K. Transformation of animal genomics by next-generation sequencing technologies: a decade of challenges and their impact on genetic architecture. Crit Rev Biotechnol 2018, 38, 1157-1175, doi:10.1080/07388551.2018.1451819.
  3. Hutchison, C.A., 3rd. DNA sequencing: bench to bedside and beyond. Nucleic acids research 2007, 35, 6227-6237, doi:10.1093/nar/gkm688.
  4. Sanger, F.; Nicklen, S.; Coulson, A.R. DNA sequencing with chain-terminating inhibitors. Proceedings of the National Academy of Sciences of the United States of America 1977, 74, 5463-5467, doi:10.1073/pnas.74.12.5463.
  5. Pei, S.; Liu, T.; Ren, X.; Li, W.; Chen, C.; Xie, Z. Benchmarking variant callers in next-generation and third-generation sequencing analysis. Brief Bioinform 2021, 22, bbaa148, doi:10.1093/bib/bbaa148.
  6. Liu, L.; Li, Y.; Li, S.; Hu, N.; He, Y.; Pong, R.; Lin, D.; Lu, L.; Law, M. Comparison of next-generation sequencing systems. J Biomed Biotechnol 2012, 2012, 251364, doi:10.1155/2012/251364.
  7. Nurk, S.; Koren, S.; Rhie, A.; Rautiainen, M.; Bzikadze, A.V.; Mikheenko, A.; Vollger, M.R.; Altemose, N.; Uralsky, L.; Gershman, A.; et al. The complete sequence of a human genome. Science (New York, N.Y.) 2022, 376, 44-53, doi:10.1126/science.abj6987.
  8. Diaz, M.H.; Winchell, J.M. The evolution of advanced molecular diagnostics for the detection and characterization of Mycoplasma pneumoniae. Frontiers in microbiology 2016, 7, 232.
  9. Voelkerding, K.V.; Dames, S.A.; Durtschi, J.D. Next-generation sequencing: from basic research to diagnostics. Clinical chemistry 2009, 55, 641-658.
  10. Microbiology, A.A.o. Applications of Clinical Microbial Next‐Generation Sequencing: Report on an American Academy of Microbiology colloquium held in Washington, DC, in April 2015. 2016.
  11. Golan, D.; Medvedev, P. Using state machines to model the Ion Torrent sequencing process and to improve read error rates. Bioinformatics 2013, 29, i344-351, doi:10.1093/bioinformatics/btt212.
  12. Shendure, J.; Porreca, G.J.; Reppas, N.B.; Lin, X.; McCutcheon, J.P.; Rosenbaum, A.M.; Wang, M.D.; Zhang, K.; Mitra, R.D.; Church, G.M. Accurate multiplex polony sequencing of an evolved bacterial genome. Science (New York, N.Y.) 2005, 309, 1728-1732, doi:10.1126/science.1117389.
  13. Barba, E.; Tsermpini, E.-E.; Patrinos, G.P.; Koromina, M. Genome Informatics Pipelines and Genome Browsers. In Applied Genomics and Public Health; Elsevier: 2020; pp. 149-169.
  14. Gourle, H.; Karlsson-Lindsjo, O.; Hayer, J.; Bongcam-Rudloff, E. Simulating Illumina metagenomic data with InSilicoSeq. Bioinformatics 2019, 35, 521-522, doi:10.1093/bioinformatics/bty630.
  15. Perez-Enciso, M.; Ferretti, L. Massive parallel sequencing in animal genetics: wherefroms and wheretos. Anim Genet 2010, 41, 561-569, doi:10.1111/j.1365-2052.2010.02057.x.
  16. Goodwin, S.; McPherson, J.D.; McCombie, W.R. Coming of age: ten years of next-generation sequencing technologies. Nat Rev Genet 2016, 17, 333-351, doi:10.1038/nrg.2016.49.
  17. Feng, Y.; Zhang, Y.; Ying, C.; Wang, D.; Du, C. Nanopore-based fourth-generation DNA sequencing technology. Genomics Proteomics Bioinformatics 2015, 13, 4-16, doi:10.1016/j.gpb.2015.01.009.

Comment-5: Line 243: "desiderable genetic traits" is better than "wanted genes"

 [Response] We modified the word with “desirable genetic traits” as suggested by the reviewer (line366).

Comment-6: 3.5 section: probably you wanted to write "insight".

 [Response] We modified the word "insight"(line 369).

Comment-7: Lines 285-297: please cite at least a reference about the statement reported in the text.

 [Response] The four references are included in the section heading “Databases and Online Resources”

References

  1. Bhati, M.; Kadri, N.K.; Crysnanto, D.; Pausch, H. Assessing genomic diversity and signatures of selection in Original Braunvieh cattle using whole-genome sequencing data. BMC genomics 2020, 21, 27, doi:10.1186/s12864-020-6446-y.
  2. Rexroad, C.; Vallet, J.; Matukumalli, L.K.; Reecy, J.; Bickhart, D.; Blackburn, H.; Boggess, M.; Cheng, H.; Clutter, A.; Cockett, N.; et al. Genome to Phenome: Improving Animal Health, Production, and Well-Being - A New USDA Blueprint for Animal Genome Research 2018-2027. Front Genet 2019, 10, 327, doi:10.3389/fgene.2019.00327.
  3. Hu, R.; Yao, R.; Li, L.; Xu, Y.; Lei, B.; Tang, G.; Liang, H.; Lei, Y.; Li, C.; Li, X.; et al. A database of animal metagenomes. Sci Data 2022, 9, 312, doi:10.1038/s41597-022-01444-w.
  4. Ko, G.; Kim, P.G.; Cho, Y.; Jeong, S.; Kim, J.Y.; Kim, K.H.; Lee, H.Y.; Han, J.; Yu, N.; Ham, S.; et al. Bioinformatics services for analyzing massive genomic datasets. Genomics Inform 2020, 18, e8, doi:10.5808/GI.2020.18.1.e8.

Comment-8: Line 353-361: references are lacking. Furthermore, please implement this paragraph. Also T. gondii in an important neglected but spread and ri-emerged zoonoses, so I suggest to include it.

 [Response] The references are added in the paragraph. We are thankful to the reviewer for sharing the above-mentioned points. We have included the section 5 (lines 495-512).

[One of the more prevalent zoonotic parasite diseases is toxoplasmosis. The late 1930s saw the first recognition of T. gondii-related disease in humans. The primary mechanism of vertical transmission of T. gondii involves tachyzoites [138]. Although tachyzoites of T. gondii have been discovered in the milk of a number of intermediate hosts, including sheep, goats, and cows, however, a report suggested that acute toxoplasmosis in humans has mostly been associated with the intake of unpasteurized goat's milk [139,140]. Furthermore, it is considered that T. gondii found in livestock meat, is a significant source of infection for people [141].

Many unknown disease-related and zoonosis-causing mutations have been discovered through advances in genome sequencing [142]. The NGS sheds fresh light on the zoonotic spread of microorganisms. High-resolution or ultra-deep sequencing showed the genetic diversity of influenza A and hepatitis E [97,143]. HT-NGS techniques were utilized for the genomic sequencing of influenza (H1N1) from animals. HTS-based metagenomic methods can be utilized to investigate new etiology outbreaks such as understanding host responses to diverse viral infections, gaining information on potential well-known illnesses suspected of having a multi-factorial etiology, and epidemic control through quick diagnosis, high sensitivity, and flexible analysis. Thus, it has the potential to lead to several new advancements in food safety and public health [144].]

References

  1. Tenter, A.M.; Heckeroth, A.R.; Weiss, L.M. Toxoplasma gondii: from animals to humans. Int J Parasitol 2000, 30, 1217-1258, doi:10.1016/s0020-7519(00)00124-7.
  2. Skinner, L.J.; Timperley, A.C.; Wightman, D.; Chatterton, J.M.; Ho-Yen, D.O. Simultaneous diagnosis of toxoplasmosis in goats and goatowner's family. Scand J Infect Dis 1990, 22, 359-361, doi:10.3109/00365549009027060.
  3. Sacks, J.J.; Roberto, R.R.; Brooks, N.F. Toxoplasmosis infection associated with raw goat's milk. Jama 1982, 248, 1728-1732, doi:10.1001/jama.1982.03330140038029.
  4. Dubey, J.P.; Lindsay, D.S.; Speer, C.A. Structures of Toxoplasma gondii tachyzoites, bradyzoites, and sporozoites and biology and development of tissue cysts. Clin Microbiol Rev 1998, 11, 267-299, doi:10.1128/cmr.11.2.267.
  5. Gilchrist, C.A.; Turner, S.D.; Riley, M.F.; Petri, W.A., Jr.; Hewlett, E.L. Whole-genome sequencing in outbreak analysis. Clin Microbiol Rev 2015, 28, 541-563, doi:10.1128/CMR.00075-13.
  6. Sobel Leonard, A.; McClain, M.T.; Smith, G.J.; Wentworth, D.E.; Halpin, R.A.; Lin, X.; Ransier, A.; Stockwell, T.B.; Das, S.R.; Gilbert, A.S.; et al. Deep Sequencing of Influenza A Virus from a Human Challenge Study Reveals a Selective Bottleneck and Only Limited Intrahost Genetic Diversification. J Virol 2016, 90, 11247-11258, doi:10.1128/jvi.01657-16.
  7. Imanian, B.; Donaghy, J.; Jackson, T.; Gummalla, S.; Ganesan, B.; Baker, R.C.; Henderson, M.; Butler, E.K.; Hong, Y.; Ring, B.; et al. The power, potential, benefits, and challenges of implementing high-throughput sequencing in food safety systems. npj Science of Food 2022, 6, 35, doi:10.1038/s41538-022-00150-6.

Comment-9: Phenomenon of purifying selection has to be more detailed. Some references touch and deal with this argue (such as Arcangeli et al 2022.).

 [Response] The authors thank reviewer #2 for the valuable comment. We have described the purifying selection with reference of Arcangeli C. et al. in detail (lines 534-538).

[Arcangeli et al. confirmed the presence of purifying selection in their studies, revealing that ss-RNA strand small ruminant lentiviruses (SRLVs) exhibit a high mutation rate and frequent recombination events, but the obtained value of non-synonymous (dN) and synonymous (dS) substitution (dN/dS) ratio indicated the presence of purifying selection [150].]

References

  1. Arcangeli, C.; Torricelli, M.; Sebastiani, C.; Lucarelli, D.; Ciullo, M.; Passamonti, F.; Giammarioli, M.; Biagetti, M. Genetic Characterization of Small Ruminant Lentiviruses (SRLVs) Circulating in Naturally Infected Sheep in Central Italy. Viruses 2022, 14, 686.

Comment-10: I suggest to mention Covid/Sars Cov2 in pandemic era when you referred to spillover question.

 [Response] Thanks reviewer #2 comment. We have mentioned COVID/SARS-CoV-2 in this section (lines 443-452).

[The severe acute respiratory syndrome coronavirus 2 (SARS-CoV-2) has been debated as either a zoonotic disease or an emerging infectious disease [97]. The COVID-19 pandemic has brought to public attention that even the highly developed and most qualified healthcare networks worldwide collapse when confronting a previously novel viral infectious disease of zoonotic origin. Before the COVID-19 pandemic, African swine fever significantly impacted the global livestock industry [98]. Following that, the COVID-19 pandemic has substantially influenced human health and the economy. The impact of the pandemic has also jeopardized the sustainability of livestock and Agri base products, significantly affecting the quality of life and economic losses. At the same time, more than 150 enteric viruses now recognized as crucial to human and animal health are considered in genomic surveillance efforts to monitor and forecast the subsequent pandemic spillover. In order to minimize cattle economic losses, advanced procedures must be prepared. Public health care considerations must also be used [99].]

References

  1. Suminda, G.G.D.; Bhandari, S.; Won, Y.; Goutam, U.; Kanth Pulicherla, K.; Son, Y.-O.; Ghosh, M. High-throughput sequencing technologies in the detection of livestock pathogens, diagnosis, and zoonotic surveillance. Computational and Structural Biotechnology Journal 2022, 20, 5378-5392, doi:https://doi.org/10.1016/j.csbj.2022.09.028.
  2. Chen, J.; Yang, C.-C. The Impact of COVID-19 on the Revenue of the Livestock Industry: A Case Study of China. Animals 2021, 11, 3586.
  3. Leifels, M.; Khalilur Rahman, O.; Sam, I.; Cheng, D.; Chua, F.J.D.; Nainani, D.; Kim, S.Y.; Ng, W.J.; Kwok, W.C.; Sirikanchana, K. The one health perspective to improve environmental surveillance of zoonotic viruses: lessons from COVID-19 and outlook beyond. Isme Communications 2022, 2, 1-9.

Comment-11: Line 396: the sentence is incomplete.

 [Response] The sentence is a subtitle. We corrected it (line 564).

Round 2

Reviewer 1 Report

After the author's revision, some useful information has been added and the manuscript has been greatly improved.

The authors have addressed my concerns. Thanks!